# AGUVIS: Unified Pure Vision Agents for Autonomous GUI Interaction

**Yiheng Xu** [* 1]  **Zekun Wang** [* 1]  **Junli Wang** [* 1]  **Dunjie Lu** [1]  **Tianbao Xie** [1]  **Amrita Saha** [2]  **Doyen Sahoo** [2]
**Tao Yu** [1]  **Caiming Xiong** [2]

## Abstract

Automating GUI tasks remains challenging due to reliance on textual representations, platform-specific action spaces, and limited reasoning capabilities. We introduce AGUVIS, a unified vision-based framework for autonomous GUI agents that directly operates on screen images, standardizes cross-platform interactions and incorporates structured reasoning via inner monologue. To enable this, we construct AGUVIS DATA COLLECTION, a large-scale dataset with multimodal grounding and reasoning annotations, and develop a two-stage training pipeline that separates GUI grounding from planning and reasoning. Experiments show that AGUVIS achieves state-of-the-art performance across offline and real-world online benchmarks, marking the first fully autonomous vision-based GUI agent that operates without closed-source models. We open-source all datasets, models, and training recipes at `https://aguvis-project.github.io` to advance future research.

## 1. Introduction

Graphical User Interfaces (GUIs) represent the primary medium of human-computer interaction in digital environments, from websites to desktop and mobile applications (Deng et al., 2023; Zhou et al., 2024; Xie et al., 2024; Rawles et al., 2024b). Creating autonomous agents that can effectively navigate these interfaces could revolutionize human productivity by enabling automated task execution using existing human-centric tools. Such automation requires mastery of three core competencies: visual understanding to comprehend complex interfaces, grounding to map natural language instructions to visual elements, and planning & reasoning to synthesize observations into effective actions. While recent advances in vision-language models (VLMs) have significantly enhanced visual interface interpretation, developing truly autonomous GUI agents remains challenging due to fundamental limitations in current approaches.

Although recent advances in large vision-language models (LVLMs) (OpenAI, 2024; Team et al., 2024; Li et al., 2024a; Wang et al., 2024b; Deitke et al., 2024; Chen et al., 2024b) have significantly enhanced the ability to interpret complex visual interfaces, we identify several critical barriers to advancing GUI automation. First, existing approaches predominantly rely on textual representations (e.g., HTML or accessibility trees) rather than visual ones (Gur et al., 2024; Kim et al., 2023; Deng et al., 2023; Zhou et al., 2024; Xie et al., 2024), whose input observation is lengthy (more than 4K), and the length increases as the complexity of the GUI grows (Xie et al., 2024), limiting generalization and increasing computational overhead compared to more natural image-based representations. Second, the heterogeneous action spaces across different platforms prevent effective cross-environment learning, constraining the available training data for each environment and impeding further scalability. Third, current methods either lack reliable visual grounding (Zheng et al., 2024a) or depend heavily on closed-source language models for reasoning (Gou et al., 2025; Lu et al., 2024), creating a fundamental bottleneck in advancing model capabilities through training. Fourth, existing methods typically train agents to generate "reactive" low-level actions directly (Hong et al., 2024; Cheng et al., 2024), failing to leverage the sophisticated reasoning capabilities inherent in vision-language models. This reactive approach struggles with complex scenarios in the real world that require careful planning and broad generalization. These limitations have prevented the development of scalable, generalizable GUI agents that can operate autonomously across diverse digital environments.

To address these challenges, we introduce AGUVIS (as shown in Figure 1), a unified vision-based framework that harmonizes visual observation and consistent action spaces across diverse GUI environments. Our approach eliminates dependence on platform-specific textual representations by operating directly on screen images, enabling more natural and generalizable interface understanding. We develop

---

[*]Equal contribution [1]University of Hong Kong [2]Salesforce Research. Correspondence to: Yiheng Xu <yhxu@cs.hku.hk>, Tao Yu <tyu@cs.hku.hk>, Caiming Xiong <cxiong@salesforce.com>.

*Proceedings of the 42$^{nd}$ International Conference on Machine Learning*, Vancouver, Canada. PMLR 267, 2025. Copyright 2025 by the author(s).

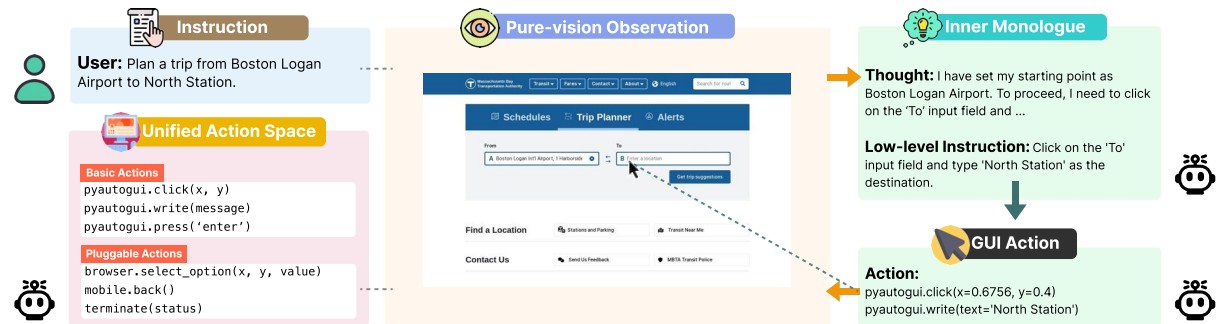

Figure 1: Overview of AGUVIS unified GUI interaction framework for autonomous GUI agents.

a standardized action space through a plugin system that maintains consistent interaction patterns across platforms while accommodating environment-specific requirements. Crucially, we incorporate explicit inner monologue during training, allowing the model to develop sophisticated reasoning patterns that emulate human problem-solving processes. This inner monologue enables the model to break down complex tasks into manageable steps, consider alternative approaches, and adapt to novel situations—capabilities that go beyond simple reactive behaviors.

To enable this unified framework, we make several technical contributions. First, we construct AGUVIS DATA COLLEC-TION, a large-scale cross-platform dataset of GUI agent trajectories that features comprehensive multimodal grounding and reasoning annotations, including explicit reasoning paths captured through inner monologue. Second, we develop a novel two-stage training pipeline that separates GUI grounding from planning and reasoning, incorporating structured thought processes to enhance autonomous navigation capabilities. Finally, we demonstrate that AGUVIS achieves state-of-the-art performance in both offline evaluation and real-world online scenarios, marking the first fully autonomous vision-based GUI agent that operates without relying on closed-source models. By open-sourcing our datasets, models, and training recipes, we provide a foundation for future research in autonomous GUI interaction.

## 2. AGUVIS

### 2.1. Problem Formulation

GUI interaction presents unique challenges due to partial observability and sequential decision-making, naturally lending itself to modeling as a Partially Observable Markov Decision Process (POMDP). We formalize this as a tuple $(\mathcal{S}, \mathcal{A}, \mathcal{O}, T, O)$, where $\mathcal{S}$ represents possible environment states, $\mathcal{A}$ denotes available actions, and $\mathcal{O}$ refers to possible observations. The state transition function $T : \mathcal{S} \times \mathcal{A} \times \mathcal{S} \rightarrow [0, 1]$ defines state transition probabilities given actions, while the observation function

$O : \mathcal{S} \times \mathcal{A} \times \mathcal{O} \rightarrow [0, 1]$ specifies observation probabilities given states and actions.

At each time step $t$, the agent receives an image observation $o_t$ from the GUI environment and generates an action $a_t$ through a structured reasoning process. This process involves inner monologue (Huang et al., 2022), which helps the agent interpret observations and determine appropriate actions. The agent then executes $a_t$, receives a new observation $o_{t+1}$, and continues until achieving the goal $G$ or reaching a terminal state.

### 2.2. Unified GUI Interaction Framework

Contemporary GUI agents predominantly rely on platform-specific representations like HTML or accessibility trees for interface interpretation, leading to fragmented approaches across different environments. We propose a unified framework that operates purely through visual observations and standardized interactions, addressing key limitations of existing methods while improving computational efficiency.

Our framework unifies both observation and action spaces across platforms while incorporating structured reasoning processes. For observations, we leverage direct visual input instead of parsing platform-specific interface code, enabling the model to process GUIs as humans do - through visual perception. The vision-centric approach not only enhances generalization across platforms but also significantly reduces computational overhead. While traditional textual methods typically require processing 4k-6k tokens per interaction—as shown in Figure 3 for HTML and reported in Xie et al. (2024) for accessibility tree—our visual approach maintains a constant token cost of 1,196 tokens for 720p images, independent of interface complexity.

At each interaction step, the agent employs a two-component inner monologue to bridge visual perception with action execution. The first component performs explicit reasoning ($h_t$) about the current state relative to the task goal $G$ and previous thoughts $h_{t-1}$, enabling adaptive planning. Finally, the agent generates precise action instruc-

tions ($a_t^{\text{instr}}$) that translate high-level intentions into concrete interface interactions. This structured thought process enables reliable handling of complex multi-step tasks.

For action execution, we adopt `pyautogui` as our universal interaction interface, supplemented by a flexible plugin system. The `pyautogui` library provides a comprehensive set of programmatic commands that mirror human input behaviors, allowing us to represent GUI interactions consistently across platforms. As shown in Table 9, this standardized action space enables the model to translate its inner monologue into concrete actions without requiring environment-specific design.

Our plugin system extends the base `pyautogui` action space to handle platform-specific requirements while maintaining the natural flow of thought-to-action conversion. It incorporates specialized interactions like mobile gestures, platform-specific shortcuts, and meta-actions such as providing responses or signaling task completion. The system aligns new actions with existing commands where possible, using explicit descriptions only when necessary, ensuring that the agent's inner reasoning remains coherent across different interaction modes. Details of these pluggable functions, particularly for mobile environments, are provided in Appendix A.2, where we describe specific mobile interaction functions and their corresponding prompts.

By integrating visual perception, structured reasoning through inner monologue, and unified action representation, our framework enables training a single model capable of operating across diverse GUI environments. This integrated approach not only simplifies the training process but also promotes human-like interaction patterns and better generalization to novel interfaces. The reduced computational overhead of visual processing, coupled with the power of structured reasoning, makes this framework particularly effective for real-world applications.

### 2.3. AGUVIS DATA COLLECTION

The effectiveness of GUI agents critically depends on high-quality training data that captures both grounding accuracy and complex reasoning patterns. However, collecting such data presents unique challenges due to the diverse nature of GUI environments and the need for detailed reasoning annotations. We address these challenges through a two-pronged data collection strategy that leverages existing resources and automated augmentation techniques shown in Figure 2. Our dataset consists of two splits: a grounding split focusing on element localization and interaction (Table 10), and a planning & reasoning split capturing multi-step task completion (Table 11). This division aligns with our framework's dual emphasis on visual understanding and structured reasoning.

**Template-augmented Grounding Data.** To create comprehensive grounding data, we employ a dual-source approach. First, we unify existing GUI datasets across platforms by converting their instruction-action annotations into our standardized `pyautogui` format. Second, we leverage the rich metadata available in broader UI datasets without action annotations, including all element positions and attributes, to generate synthetic instruction-action pairs through carefully designed templates. This approach not only expands our training data but also ensures coverage of diverse interface patterns and interaction types.

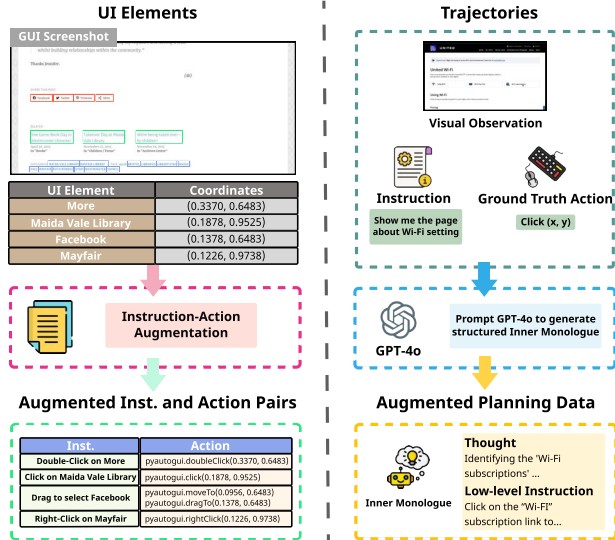

Figure 2: AGUVIS DATA COLLECTION augmentation pipeline for two-stage training data.

**VLM-augmented Planning & Reasoning Trajectories.** While existing GUI agent datasets provide high-level goals and action sequences (Deng et al., 2023; Rawles et al., 2024b; Li et al., 2024c), they often lack the intermediate reasoning steps crucial for advanced agent behavior. We address this limitation through a novel VLM-based trajectory augmentation process. For each trajectory step, we construct a rich training example by first highlighting relevant UI elements in the observation $o_t$ to guide the VLM's attention. Given the high-level goal $G$, the current image observation $o_t$, and the grounded action $a_t$, we prompt GPT-4o to generate the inner monologue components: thoughts $h_t$, and low-level action instruction $a_t^{\text{instr}}$. To maintain temporal coherence, we include previous action instructions $a_1^{\text{instr}}, \ldots, a_{t-1}^{\text{instr}}$ in the context. The complete prompting template and example are detailed in Appendix B.2 and Figure 6. Our carefully crafted approach ensures the generated inner monologues are predictive rather than post-hoc explanations, enabling the agent to develop planning capabilities. Through extensive human evaluation with quantitative analysis detailed in Appendix B.3, we validate the quality of these augmented trajectories, confirming that they effectively capture not just the actions to take, but also the complete reasoning

Table 1: Comparison of various planners and grounding methods on ScreenSpot across various device and input modalities. The top part of table shows the results on *original instructions* evaluation setting while the bottom part shows results on *self-plan* evaluation setting. Best results are in bold.

| Planner | Grounder | Mobile | | Desktop | | Web | | Avg |
|---|---|---|---|---|---|---|---|---|
| | | Text | Icon/Widget | Text | Icon/Widget | Text | Icon/Widget | |
| - | GPT-4 | 22.6 | 24.5 | 20.2 | 11.8 | 9.2 | 8.8 | 16.2 |
| | GPT-4o | 20.2 | 24.9 | 21.1 | 23.6 | 12.2 | 7.8 | 18.3 |
| | CogAgent | 67.0 | 24.0 | 74.2 | 20.0 | 70.4 | 28.6 | 47.4 |
| | SeeClick | 78.0 | 52.0 | 72.2 | 30.0 | 55.7 | 32.5 | 53.4 |
| | Qwen2-VL | 75.5 | 60.7 | 76.3 | 54.3 | 35.2 | 25.7 | 55.3 |
| | UGround | 82.8 | 60.3 | 82.5 | 63.6 | 80.4 | 70.4 | 73.3 |
| | AGUVIS-G-7B | **88.3** | **78.2** | **88.1** | **70.7** | **85.7** | **74.8** | **81.8** |
| GPT-4 | SeeClick | 76.6 | 55.5 | 68.0 | 28.6 | 40.9 | 23.3 | 48.8 |
| | OmniParser | 93.9 | 57.0 | 91.3 | 63.6 | 81.3 | 51.0 | 73.0 |
| | UGround | 90.1 | 70.3 | 87.1 | 55.7 | 85.7 | 64.6 | 75.6 |
| GPT-4o | SeeClick | 81.0 | 59.8 | 69.6 | 33.6 | 43.9 | 26.2 | 52.3 |
| | UGround | 93.4 | 76.9 | 92.8 | 67.9 | 88.7 | 68.9 | 81.4 |
| | AGUVIS-7B | **95.6** | 77.7 | 93.8 | 67.1 | 88.3 | 75.2 | **84.4** |
| | AGUVIS-72B | 94.5 | **85.2** | **95.4** | **77.9** | **91.3** | **85.9** | **89.2** |

process leading to those actions. Our analysis reveals that 86.7% of the augmented data successfully demonstrates intermediate reasoning that aligns with both the ground truth actions and the overall goal intention, with detailed failure case analysis provided in Appendix B.3.2.

## 2.4. Model Architecture

Vision-based GUI agents require direct mapping between visual observations and actions, necessitating an architecture optimized for high-resolution image processing while preserving spatial relationships. We selected Qwen2-VL as our foundation, leveraging its NaViT-style image encoder's native support for dynamic resolution processing—a critical feature for handling diverse interface layouts. Another key strength of the architecture lies in its position embedding mechanism. By replacing traditional absolute position embeddings with 2D-RoPE, the model maintains precise spatial awareness across varying screen dimensions while efficiently converting interface screenshots into visual tokens. These improvements significantly reduce computational overhead compared to conventional methods.

To validate our framework's flexibility, we also implemented it using LLaVA-OneVision, which similarly supports high-resolution image processing with variable aspect ratios, albeit with higher token costs. This implementation confirms our framework's model-agnostic nature, with detailed comparisons presented in Section 4.1.

## 2.5. Training Paradigm

The training process of AGUVIS is divided into two stages: Grounding Training and Planning & Reasoning Training.

Each stage leverages a distinct split from AGUVIS DATA COLLECTION to progressively build the agentic abilities. The complete training example templates and prompt formats for both stages are detailed in Appendix C.1.

**Stage 1: Grounding Training**  The first stage focuses on developing fundamental GUI interaction capabilities through efficient processing of single-screenshot environments. To address the challenge of multiple interactable objects within each screenshot generating redundant training data, we implement a grounding packing strategy. This approach bundles multiple instruction-action pairs into a single image, creating a single-image-multiple-turn format. By processing several grounding examples simultaneously from each screenshot, we significantly reduce training overhead while maintaining performance (shown in Appendix C.2). The result of this stage is AGUVIS-G, a model equipped with advanced GUI understanding and interaction capabilities.

**Stage 2: Planning & Reasoning Training**  Building on AGUVIS-G's foundation, the second stage develops advanced decision-making and reasoning abilities necessary for complex, multi-step tasks. We leverage our detailed inner-monologue trajectory data (as described in Section 2.3) to implement a reasoning mixture approach, exposing the model to varying levels of cognitive complexity. This ranges from basic action instructions to comprehensive inner monologues encompassing thoughts, and detailed action plans. The dynamic adjustment of trajectory complexity ensures the model develops adaptable reasoning patterns and sophisticated decision-making capabilities. The final result is AGUVIS, a fully-trained model capable of handling both offline and online GUI tasks across diverse environ-

Table 2: Performance comparison on Multimodal Mind2Web across different settings. We report element accuracy (Ele.Acc), Operation F1 (Op.F1), and step success rate (Step SR). Best results are in bold. "T" means the textual HTML code as inputs. "I" means the GUI images as inputs. More explanation about result source in Appendix D.2

| Obs. | Planner | Grounder | Cross-Task | | | Cross-Website | | | Cross-Domain | | |
|---|---|---|---|---|---|---|---|---|---|---|---|
| | | | Ele.Acc | Op.F1 | Step SR | Ele.Acc | Op.F1 | Step SR | Ele.Acc | Op.F1 | Step SR |
| T | GPT-3.5 | Choice | 19.4 | 59.2 | 16.8 | 14.9 | 56.5 | 14.1 | 25.2 | 57.9 | 24.1 |
| | GPT-4 | Choice | 40.8 | 63.1 | 32.3 | 30.2 | 61.0 | 27.0 | 35.4 | 61.9 | 29.7 |
| T + I | GPT-4 | Choice | 46.4 | 73.4 | 40.2 | 38.0 | 67.8 | 32.4 | 42.4 | 69.3 | 36.8 |
| | GPT-4 | SoM | 29.6 | - | 20.3 | 20.1 | - | 13.9 | 27.0 | - | 23.7 |
| I | GPT-4o | SeeClick | 32.1 | - | - | 33.1 | - | - | 33.5 | - | - |
| | GPT-4V | OmniParser | 42.4 | 87.6 | 39.4 | 41.0 | 84.8 | 36.5 | 45.5 | 85.7 | 42.0 |
| | GPT-4o | UGround | 47.7 | - | - | 46.0 | - | - | 46.6 | - | - |
| I | SeeClick-9.6B | | 28.3 | 87.0 | 25.5 | 21.4 | 80.6 | 16.4 | 23.2 | 84.8 | 20.8 |
| | AGUVIS-7B | | 64.2 | 89.8 | 60.4 | 60.7 | 88.1 | 54.6 | 60.4 | **89.2** | 56.6 |
| | AGUVIS-72B | | **69.5** | **90.8** | **64.0** | **62.6** | **88.6** | **56.5** | **63.5** | 88.5 | **58.2** |

ments with nuanced understanding and precision. Complete training implementation details, including hardware configurations, training durations, and hyperparameters, are provided in Appendix C.2.

## 3. Experiments

### 3.1. GUI Grounding Evaluation

**ScreenSpot.** We first evaluated AGUVIS's fundamental GUI grounding capabilities using ScreenSpot (Cheng et al., 2024), a benchmark that spans mobile, desktop, and website platforms. Following established protocols from previous work (Cheng et al., 2024; Gou et al., 2025), we tested under two conditions: direct execution from original instructions and self-planned execution requiring natural language planning before action.

The results in Table 1 demonstrate AGUVIS's exceptional grounding capabilities across platforms. Our AGUVIS-G-7B, trained with the proposed grounding stage, significantly outperforms existing models on original instructions. The full model AGUVIS-7B shows even stronger performance after planning trajectory training, surpassing previous approaches that rely on closed-source LLMs like GPT-4o. The scaled version, AGUVIS-72B, achieves state-of-the-art performance with an average score of 89.2.

### 3.2. Offline GUI Agent Evaluation

We assessed AGUVIS's planning capabilities through two major offline benchmarks: Multimodal-Mind2Web (Zheng et al., 2024a) for website interaction and AndroidControl (Li et al., 2024c) for mobile device operation.

**Multimodal-Mind2Web.** Multimodal-Mind2Web evaluations focused on website navigation and interaction tasks. Unlike previous approaches that utilize textual inputs (Deng

et al., 2023) or Set-of-Marks (Zheng et al., 2024a), AGUVIS operates solely on GUI screenshots. The results in Table 2 show that AGUVIS achieves superior performance across all metrics, with a particularly notable improvement in Step Success Rate (+51.9% on average), highlighting enhanced planning capabilities.

Table 3: Step Accuracy of out-of-domain data on Android-Control under high-level tasks and low-level tasks. Best performance is in bold. "Acc.Tree" means the textual accessibility tree.

| Obs. | Planner | Grounder | Step Acc. | |
|---|---|---|---|---|
| | | | High | Low |
| Acc. Tree | GPT-4-Turbo | Choice | 42.1 | 55.0 |
| | PaLM 2S* | Choice | 58.5 | 77.5 |
| Image | GPT-4-Turbo | SeeClick | 39.4 | 47.2 |
| | GPT-4-Turbo | UGround | 46.2 | 58.0 |
| | GPT-4o | SeeClick | 41.8 | 52.8 |
| | GPT-4o | UGround | 48.4 | 62.4 |
| Image | AGUVIS-7B | | **61.5** | **80.5** |
| | AGUVIS-72B | | **66.4** | **84.4** |

**AndroidControl.** For mobile interface interaction, we evaluated AGUVIS on AndroidControl using a subset of 500 randomly sampled step-actions following the setting in Li et al. (2024c). We tested both high-level planning scenarios and low-level instruction execution, comparing against models using various input modalities. Table 3 demonstrates AGUVIS's superior performance in both settings, confirming its effectiveness across different interaction paradigms.

### 3.3. Online GUI Agent Evaluation

To validate real-world applicability, we evaluated AGUVIS across three comprehensive benchmarks: Mind2Web-Live (Pan et al., 2024), AndroidWorld (Rawles et al., 2024a), MobileMiniWob (Rawles et al., 2024b).

Mind2Web-Live provides a dynamic web-based environment derived from Mind2Web, evaluating task completion through step-by-step success rates. AndroidWorld operates in a virtual Android environment, using a Pixel 6 phone simulator for mobile agent assessment. MobileMiniWob adapts 92 tasks from MiniWob++ to AndroidWorld environment, providing standardized GUI interaction scenarios. More details are in Appendix D.3. We test two distinct configurations: one pairs GPT-4o as a planner with AGUVIS-7B as a grounder, and the other employs AGUVIS-72B in both roles.

Table 4: Task Success Rate (SR) and efficiency costs on Mind2Web-Live. Cost is calculated by dividing the model's total inference cost in USD by the number of successful steps.

| Inputs | Planner | Grounder | SR | Cost |
|---|---|---|---|---|
| HTML | GPT-4-Turbo | Choice | 21.1 | - |
| | GPT-4o | Choice | 22.1 | 0.142 |
| | Llama-3.1-405B | Choice | 24.0 | 0.174 |
| | Llama-3.1-70B | Choice | 20.2 | 0.031 |
| | GPT-3.5-turbo | Choice | 17.3 | 0.092 |
| Image | GPT-4-Turbo | UGround | 23.1 | - |
| | GPT-4o | UGround | 19.2 | - |
| | GPT-4o | AGUVIS-7B | **24.0** | **0.106** |
| Image | AGUVIS-72B | | **27.1** | **0.012** |

**Results** Tables 4 and 5 present our comprehensive findings. When using GPT-4o as the planner, AGUVIS-7B demonstrates superior performance across benchmarks compared to existing methods. The unified AGUVIS-72B approach achieves best-in-class performance on Mind2Web-Live and MobileMiniWob. These results, combined with our model's significant efficiency advantages, demonstrate the strong potential of pure vision-based agents for real-world GUI automation tasks.

Table 5: Task Success Rates (SR) on AndroidWorld (AW) and MobileMiniWob(MMW). Best results are in bold.

| Input | Planner | Grounder | $AW_{SR}$ | $MMW_{SR}$ |
|---|---|---|---|---|
| AXTree | GPT-4-Turbo | Choice | 30.6 | 59.7 |
| | Gemini 1.5 Pro | Choice | 19.4 | 57.4 |
| Image + AXTree | GPT-4-Turbo | SoM | 25.4 | 67.7 |
| | Gemini 1.5 Pro | SoM | 22.8 | 40.3 |
| Image | GPT-4-Turbo | UGround | 31.0 | - |
| | GPT-4o | UGround | 32.8 | - |
| | GPT-4o | AGUVIS-7B | **37.1** | **55.0** |
| Image | AGUVIS-72B | | **26.1** | **66.0** |

## 4. Analysis

### 4.1. Impact of Training Stages

We first assess the impact of each stage in our training pipeline by evaluating several variants of AGUVIS. As shown in Table 6, we examine the performance of: (a) a model trained without the second stage (planning & reasoning), referred to as AGUVIS-G, and (b) Qwen2-VL, the base model without both stages of specialized training. The results demonstrate clear performance degradation when either training stage is omitted. In particular, removing Stage 2 (planning & reasoning) leads to significant drops in performance across all metrics.

To verify that these improvements stem from our methodology rather than the inherent capabilities of Qwen2-VL, we conducted parallel experiments using LLaVA as an alternative backbone. The results in Table 15 show that even with a weaker foundation model, the AGUVIS training pipeline yields substantial improvements. For instance, LLaVA's performance on ScreenSpot improves from 3.8% to 81.2% after applying our complete training process, validating the effectiveness of our approach across different architectures.

### 4.2. Role of Inner Monologue

Inner monologue plays a crucial role in enhancing both planning and grounding capabilities. As demonstrated in Table 6, removing inner monologue from training data results in significant performance drops across all benchmarks. The impact is particularly noticeable in low-level tasks. For instance, ScreenSpot performance falls from 84.4% to 79.3%, and low-level AndroidControl drops from 80.5% to 69.1%. This suggests that inner monologue aids not only high-level planning but also precise low-level execution. More detailed results are in Appendix E.1.2.

### 4.3. Cross-Platform Benefits

Our unified training approach enables effective knowledge transfer across different platforms, allowing the model to develop generalizable interaction capabilities. As shown in Table 7, training on both web and mobile data leads to significantly better performance compared to platform-specific training. On web-specific tasks in Multimodal-Mind2Web, models trained with both web and mobile data achieve superior results compared to those trained solely on web data or Mind2Web data alone.

These improvements highlight the ability of our framework to leverage commonalities across different GUI environments, fostering generalization beyond individual datasets. The combination of a pure vision approach and standardized `pyautogui` actions establishes a shared representation space, enabling effective cross-platform learning.

Table 6: Ablation on AGUVIS-7B on MM-Mind2Web and AndroidControl benchmarks. We report the step success rate. We provide a more comprehensive ablation in Appendix E.1

| Settings | ScreenSpot | Multimodal-Mind2Web | | | AndroidControl | |
|---|---|---|---|---|---|---|
| | | Cross-Task | Cross-Website | Cross-Domain | High-Level | Low-Level |
| AGUVIS-7B | 84.4 | 60.4 | 54.6 | 56.6 | 61.5 | 80.5 |
| (a) w/o Stage 2 | 81.8 | 50.9 | 45.2 | 45.3 | 58.0 | 75.6 |
| (b) w/o Stage 1 | 77.4 | 59.7 | 55.3 | 56.8 | 58.8 | 79.8 |
| (c) w/o Stage 1 & 2 | 55.3 | 50.9 | 44.9 | 47.7 | 59.1 | 59.2 |
| (d) w/o Inner Monologue | 79.3 | 55.4 | 53.7 | 54.9 | 60.3 | 69.1 |

Table 7: Ablation study on Multimodal-Mind2Web, analyzing the impact of training data from different device domains within a unified action space.

| Data | #Traj. | Task | Website | Domain |
|---|---|---|---|---|
| Web + Mobile | 35k | 58.5 | 55.4 | 54.8 |
| Web Only | 6k | 53.1 | 50.3 | 52.2 |
| Mind2Web Only | 1k | 50.9 | 44.9 | 47.7 |

To further evaluate the generalization capabilities of our model, we tested it on OSWorld, a unified computer environment designed for multimodal agents. OSWorld presents complex workflows, encompassing 369 real-world computer tasks that span web applications, desktop software, and OS-level operations. The results are shown in Table 8.

Remarkably, despite being trained exclusively on web and mobile trajectory data, our model demonstrates strong generalization to desktop GUI tasks. On OSWorld, when paired with GPT-4o for planning, our model achieves a 17.04% task success rate, significantly outperforming SoM-based approaches (4.59%) and even surpassing Claude Computer-Use (14.9%). Furthermore, AGUVIS-72B, when deployed as an independent model, achieves 10.26%, demonstrating that our approach is competitive even without external planning support. This result underscores that our approach does not overfit to specific environments but instead captures fundamental GUI interaction principles, enabling effective transfer to novel computing scenarios.

Table 8: Success rate on the OSWorld benchmark in the screenshot-only setting.

| Planner | Grounding | Task SR |
|---|---|---|
| GPT-4o | SoM | 4.59 |
| GPT-4o | AGUVIS-7B | 14.79 |
| GPT-4o | AGUVIS-72B | 17.04 |
| GPT-4o | | 5.03 |
| GPT-4V | | 5.26 |
| Gemini-Pro-1.5 | | 5.40 |
| Claude Computer-Use | | 14.9 |
| OpenAI Operator | | **19.7** |
| AGUVIS-72B | | 10.26 |

in Mind2Web-Live, as detailed in Figure 3 and Table 4.

Figure 3: Comparison of Input Tokens per Step and USD Efficiency in GUI Interaction. The bar chart shows the input tokens required per step during GUI interactions, while the line graph illustrates USD Efficiency for all models.

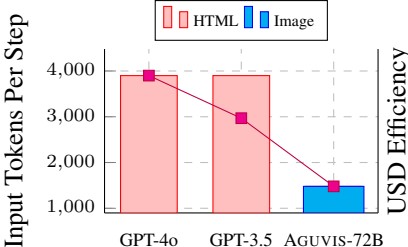

### 4.4. Efficiency Benifits from Pure Vision Perception

The pure vision approach significantly reduces computational overhead compared to traditional textual methods. As illustrated in Figure 3, while HTML-based approaches typically require processing about 4,000 tokens per interaction, our vision-based method maintains a constant token cost of 1,196 tokens for 720p images, independent of interface complexity. This efficiency translates to substantial practical benefits in deployment, with our method reducing costs by 93% and input tokens per step by 70% compared to GPT-4o

### 4.5. Error Analysis and Future Work

To understand failure modes and potential improvements, we conducted a detailed error analysis on 50 samples from the ScreenSpot dataset under the self-plan setting. Our analysis reveals two primary categories of errors, as shown in Figure 4: 40% stem from ambiguous instructions that could refer to multiple grounding targets, while the remaining 60% are grounding errors. A critical finding is that the model currently lacks the ability to indicate uncertainty or refuse actions when faced with ambiguous instructions - an essen-

tial capability for real-world deployment where incorrect actions could have significant consequences.

Figure 4: Error analysis on Screenspot under the self-plan setting.

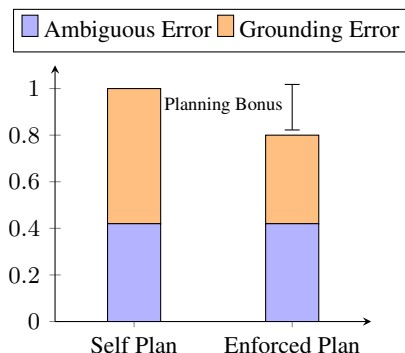

When we enforce planning by prompting the agent model to generate inner monologue before execution - as detailed in Appendix E.2.1 - it resolves 20% of these grounding errors, suggesting that explicit reasoning helps the model leverage its knowledge more effectively. However, our analysis reveals a significant challenge: while many queries appear simple syntactically, they actually require deeper semantic understanding and domain knowledge. In these cases, the model struggles to recognize the need for planning and defaults to direct grounding instead of explicit reasoning. We provide illustrative examples of these semantically challenging cases in Appendix E.2.2.

These insights highlight several promising directions for future work, along with concrete solutions. To develop more reliable GUI agents for real-world settings, we propose incorporating adversarial training examples where the correct action is to refuse execution or raise safety concerns, helping models learn to handle ambiguous or potentially harmful situations appropriately. To enhance the model's ability to identify semantically complex tasks requiring planning, we suggest augmenting training data with explicit annotations of task complexity and required reasoning depth, potentially combined with a dynamic threshold system during inference to balance planning overhead against accuracy gains. By pursuing these directions, we can work toward GUI agents that not only perform tasks accurately but also do so with appropriate caution and self-awareness - a crucial requirement for real-world deployment.

## 5. Related Work

### 5.1. GUI Agent Benchmarks

Recent advancements in autonomous GUI agents have spurred the development of numerous benchmarks assessing agent capabilities across diverse platforms, including those focused on the Web (Deng et al., 2023; Zhou et al.,

2024; Koh et al., 2024a; Lù et al., 2024; Drouin et al., 2024; Pan et al., 2024), desktop (Xie et al., 2024; Bonatti et al., 2024), and mobile environments (Rawles et al., 2024a;b; Zhang et al., 2024b; Chai et al., 2024; Lu et al., 2024; Li et al., 2024c). Furthermore, cross-platform datasets such as ScreenSpot (Cheng et al., 2024), OmniACT (Kapoor et al., 2024), GUICourse (Chen et al., 2024a), and CRAB (Xu et al., 2024a) provide comprehensive evaluation frameworks spanning multiple devices and interfaces. Evaluations on specialized applications have also emerged, such as Spider-2V (Cao et al., 2024) targeting data science and engineering workflows. To thoroughly evaluate our proposed model's grounding and planning capabilities, we conduct extensive experiments on relevant benchmarks under both online and offline settings.

### 5.2. GUI Agent Models

Significant progress has been made in developing more capable autonomous GUI agents. For web navigation, models such as WebGPT (Nakano et al., 2021), Lemur (Xu et al., 2024b), Agent-Lumos (Yin et al., 2024), CogAgent (Hong et al., 2024), AutoWebGLM (Lai et al., 2024) and xLAM (Zhang et al., 2024a) have demonstrated enhanced performance. Auto-GUI (Zhang & Zhang, 2024), AppAgent (Zhang et al., 2025), and ScreenAgent (Niu et al., 2024) propose novel approaches for direct GUI interaction without relying on application-specific APIs. More recently, research has targeted core capabilities of GUI agents like grounding and planning & reasoning. Gou et al. (2025) and Wu et al. (2025) propose scaling grounding data to improve grounding ability. As for enhancing the planning and reasoning ability of GUI agents, some GUI agents for mobile environments (Li et al., 2023; 2024b; Wang et al., 2024a) explicitly incorporate planning trajectories for training. Koh et al. (2024b) introduces an inference-time search algorithm in interactive web environments. These advancements collectively enable more sophisticated and capable GUI agents for automated task completion across digital platforms.

## 6. Conclusion

We introduced AGUVIS, a unified pure vision-based framework for autonomous GUI agents that operate across diverse platforms. By leveraging vision-only observations and a standardized action space, AGUVIS eliminates reliance on platform-specific representations and closed-source models. Our structured reasoning approach, combined with a large-scale dataset and a two-stage training pipeline, enables superior grounding, planning, and reasoning. Extensive experiments demonstrate state-of-the-art performance in both offline and online GUI tasks. We open-source all datasets, models, and training recipes to accelerate future research in this domain.

## Acknowledgment

This project is partially funded by the ECS (27212023) from the Hong Kong RGC.

## Impact Statement

Creating autonomous agents capable of navigating Graphical User Interfaces (GUIs) has the potential to revolutionize human productivity by automating tasks using existing human-centric tools. In this work, we focus on democratizing GUI agents by leveraging open-source models, rather than relying on proprietary LLMs. By training open-source models to possess native planning and reasoning capabilities, we enable, for the first time, end-to-end autonomous GUI interactions based on pure vision and open-source models. Security remains paramount in real-world applications of GUI agents. Ensuring that agents do not execute harmful actions is a critical area that warrants further investigation. To foster the advancement of future research on GUI agents and computer-use tasks, we have open-sourced all relevant datasets, models, and training recipes. This initiative aims to provide a foundational platform for researchers and developers to build upon, encouraging innovation and collaboration in the field.

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

# Table of Contents in Appendix

## A. AGUVIS Unified Design

### A.1. Details of Action Space in AGUVIS

In this section, we introduce our unified action space of our pure vision agent framework AGUVIS. As shown in Table 9, we use default standard `pyautogui` actions with pluggable actions as the action space of AGUVIS, which ensures the agent model's universality across environments as well as its flexibility in the specific environment.

Table 9: Default standard `pyautogui` actions $\mathcal{A}$ with pluggable actions.

| Category | Action Space |
| --- | --- |
| Basic Actions | `pyautogui.moveTo(x, y)` 
 `pyautogui.click(x, y)` 
 `pyautogui.write('text')` 
 `pyautogui.press('enter')` 
 `pyautogui.hotkey('ctrl', 'c')` 
 `pyautogui.scroll(200)` 
 `pyautogui.dragTo(x, y)` |
| Pluggable Actions 
 ... | `browser.select_option(x, y, value)` 
 `mobile.swipe(from, to)` 
 `mobile.home()` 
 `mobile.back()` 
 `mobile.open_app(name)` 
 `terminate(status)` 
 `answer(text)` 
 ... |

### A.2. Pluggable Functions: Mobile Environments as An Example

We provide the following pluggable functions for Aguvis in the mobile environment, along with their corresponding descriptions.

```
Pluggable Functions for AGUVIS

You are a GUI agent. You are given a task and a screenshot of the screen. You need
to perform a series of pyautogui actions to complete the task.

You have access to the following functions:
- {"name": "mobile.home", "description": "Press the home button"}
- {"name": "mobile.back", "description": "Press the back button"}
- {
    "name": "mobile.long_press",
    "description": "Long press on the screen",
    "parameters": {
        "type": "object",
        "properties": {"x": {"type": "number", "description": "The x coordinate of
        the long press"}, "y": {"type": "number", "description": "The y coordinate
        of the long press"}},
        "required": ["x", "y"]
    }
  }
- {
    "name": "mobile.open_app",
    "description": "Open an app on the device",
    "parameters": {
        "type": "object",
        "properties": {"app_name": {"type": "string", "description": "The name of
        the app to open"}},
        "required": ["app_name"]
    }
  }
- {
    "name": "terminate",
    "description": "Terminate the current task and report its completion status",
    "parameters": {
        "type": "object",
        "properties": {"status": {"type": "string", "enum": ["success"],
        "description": "The status of the task"}},
        "required": ["status"]
    }
  }
- {
    "name": "answer",
    "description": "Answer a question", "parameters": {
        "type": "object",
        "properties": {"answer": {"type": "string", "description": "The answer to
        the question"}},
        "required": ["answer"]
    }
  }
```

# B. Data Curation of AGUVIS DATA COLLECTION

### B.1. Detailed Source Dataset Statistics

We present the detailed statistical information of all training datasets utilized in both the grounding and planning & reasoning stages. The statistics are shown in Table 10 and Table 11, respectively.

Table 10: The grounding split of AGUVIS DATA COLLECTION. Each example in this split consists of a single-step trajectory.

| Data source | Platform | Instruction | #Trajectory |
|---|---|---|---|
| SeeClick (Cheng et al., 2024) | Website | Augmented | 271K |
| GUIEnv (Chen et al., 2024a) | Website | Augmented | 328K |
| GUIAct (Chen et al., 2024a) | Website | Original | 67K |
| WebUI (Wu et al., 2023) | Website | Augmented | 57K |
| Widget Captioning (Li et al., 2020b) | Mobile | Original | 101K |
| RicoSCA (Li et al., 2020a) | Mobile | Original | 173K |
| UI RefExp (Bai et al., 2021) | Mobile | Original | 16K |
| RICO Icon (Deka et al., 2017) | Mobile | Augmented | 16K |
| OmniACT (Kapoor et al., 2024) | Desktop & Website | Original | 7K |
| Total | | | 1.036M |

## B.2. Prompt for Augmenting Planning & Reasoning Trajectories

**Prompt for GPT-4o generating planning & reasoning data**

```
Goal: {goal}
Previous Actions: {previous_actions}

Given the current screenshot and the next ground truth action labeled as
`{current_action_instruction}`, the action commands is:
```json
{action_commands}
```
This element is highlighted in red bounding box in the image.

Describe the situation in detail, focusing on the goal and current observation.
Ensure your reasoning aligns with the goal and the labeled action, but avoid using
the labeled action or the highlighted bounding box as reasoning support, as they
represent hindsight rather than predictive insight. Conclude with a clear,
actionable instruction in one sentence. Aim to reason through the task as if solving
it, rather than simply reflecting on the labeled outcome. Use the first-person
perspective to represent the annotator's thought process.
```

We use GPT-4o as the foundational model to augment our integrated agent trajectory. In this stage, the `Goal` represents the target of the trajectory, `Previous Actions` is a stack of all previous low-level instructions, `current_action_instruction` refers to the low-level instruction corresponding to the current action in the dataset, and `action_commands` is the representation of the current action in the form of `pyautogui` code within the dataset. We show the augmented examples generated by GPT-4o in Figure 5. This augmentation data serves to enrich reasoning trajectories and can be generated using open-source VLMs (Bai et al., 2025), we leave exploring that approach for future work.

Table 11: The planning & reasoning split of AGUVIS DATA COLLECTION.

| Data source | Platform | Inner Monologue | Avg. Steps | #Trajectory |
|---|---|---|---|---|
| MM-Mind2Web (Zheng et al., 2024a) | Website | Generated | 7.7 | 1,009 |
| GUIAct (Chen et al., 2024a) | Website | Generated | 6.7 | 2,482 |
| MiniWoB++ (Zheng et al., 2024b) | Website | Generated | 3.6 | 2,762 |
| AitZ (Zhang et al., 2024b) | Mobile | Original | 6.0 | 1,987 |
| AndroidControl (Li et al., 2024c) | Mobile | Original | 5.5 | 13,594 |
| GUI Odyssey (Lu et al., 2024) | Mobile | Generated | 15.3 | 7,735 |
| AMEX (Chai et al., 2024) | Mobile | Generated | 11.9 | 2,991 |
| AitW (Rawles et al., 2024b) | Mobile | Generated | 8.1 | 2,346 |
| Total | | | | 35K |

## B.3. Human Study on Augmented Data

### B.3.1. QUALITATIVE HUMAN STUDY

Based on our findings that our Augmented Planning and Reasoning Data improves the performance of Aguvis, we conducted a qualitative study on augmented data. From the VLM-augmented data, we selected 90 samples for a human study and evaluated them according to specific criteria.

We determined that for augmented data to be considered successful, it must:

- Match the action type and action target elements of the ground truth,

- Correctly describe the step's intention,

- Establish a clear connection between the step's intention and the overall goal,

- Assist the agent in successfully completing the task.

Among the sampled data, we found that 86.7% demonstrated intermediate reasoning that aligned with the ground truth actions and the overall goal's action intention. The remaining 7.8% cases were influenced by dataset noise (irrelevant or unnecessary actions within the task), and 5.5% cases were due to misinterpretations of the action intention under clean data.

### B.3.2. FAILURE CASES UNDER NOISY TRAINING DATA

We analyzed error cases in the generated data and identified several issues. Specifically, we found that unnecessary actions in the training data can lead to the VLM failing to establish a connection between these extra actions and the overall goal, ultimately resulting in incorrect reasoning and planning.

While these redundant actions do not compromise the trajectory's overall completeness or correctness, they do introduce challenges for the VLM in generating accurate planning.

# C. AGUVIS Training

## C.1. Training Example Schema

---

**Training Data Schema of Stage 1 Grounding**

**Prompt**

```
<|im_start|>system
You are a GUI agent. You are given a task and a screenshot of the screen. You need
to perform a series of pyautogui actions to complete the task.<|im_end|>
<|im_start|>user
<|vision_start|><|image_pad|><|vision_end|>
Please generate the next move according to the ui screenshot, instruction and
previous actions.
Instruction: {overall_goal}
Previous actions: {previous_actions}
<|im_end|>
```

- - - - - - - - - - - - - - - - - - - - - - - - - - - - - - - - - - - - - - - - - -

**Generation**

```
<|im_start|>assistant<|recipient|>os
Action: {pyautogui function}
<|diff_marker|>
```

---

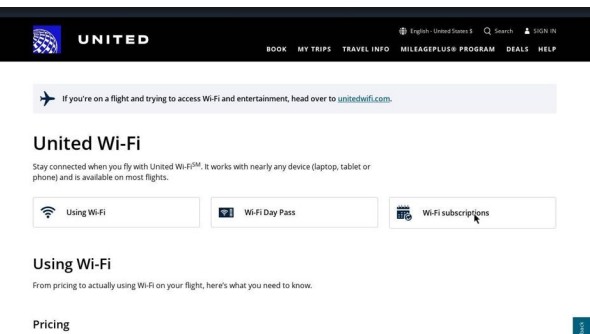

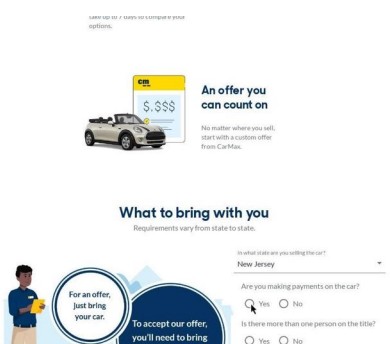

**Goal:** Show me the page with information about Wi-Fi subscriptions.

**Goal:** Find the documents required to sell a car in New Jersey for which payments are being paid and it has single person on the title who can come to the store

**Previous actions:**
Step 1: Click on the 'TRAVEL INFO' tab to explore available options for Wi-Fi subscription information.
Step 2: Click on the 'Pre-paid Wi-Fi' link to access Wi-Fi subscription information.

**Previous actions:**
Step 1: Click on the 'Sell/Trade' link in the navigation bar to proceed.
Step 2: Select 'New Jersey' from the dropdown menu labeled 'In what state are you selling the car?'.

**Action:** pyautogui.click(x=0.8018, y=0.6183)

**Action:** pyautogui.click(x=0.5938, y=0.8633)

### Generated Planning

### Generated Planning

**Thought:** Identifying the 'Wi-Fi subscriptions' link on the United Wi-Fi page aligns perfectly with the goal of finding subscription information. I am considering available options and see a dedicated link for subscriptions.

**Thought:** To determine the document requirements for selling a car in New Jersey, I need to select options that accurately reflect the current status of the car's payments and ownership. Since payments are being made, I will choose 'Yes' for that question to proceed and assess any further requirements or conditions this might bring into play.

**Low-level Instruction:** Click on the 'Wi-Fi subscriptions' link to access information about Wi-Fi subscriptions.

**Low-level Instruction:** Click on the 'Yes' radio button next to 'Are you making payments on the car?'

Figure 5: Examples of augmented planning and reasoning data generated by GPT-4o. The position of the mouse in the image represents the ground truth click position in the training data.

---

**Training Data Schema of Stage 2 Planning**

**Prompt**

```
<|im_start|>system
You are a GUI agent. You are given a task and a screenshot of the screen. You need
to perform a series of pyautogui actions to complete the task.<|im_end|>
<|im_start|>user
<|vision_start|><|image_pad|><|vision_end|>
Please generate the next move according to the ui screenshot, instruction and
previous actions.
Instruction: {overall_goal}
Previous actions: {previous_actions}
<|im_end|>
```

**Generation**

```
<|im_start|>assistant<|recipient|>all
Thought: {Planning}
Low-level Instruction: {Low-level Instruction}
<|im_end|>
<|im_start|>assistant<|recipient|>os
Action: {pyautogui function}
<|diff_marker|>
```

---

AGUVIS introduces a novel explicit planning and reasoning training framework that differs from existing approaches. We illustrate these differences with visual examples in Figure 6. While existing training datasets utilize trajectory data to fine-tune agents, these approaches often involve agents directly outputting action commands (e.g., via pyautogui), bypassing the generation of observations, thoughts, and low-level instructions in natural language that correspond to actions. To elicit the reasoning and planning capabilities of vision-language models and provide the model with richer context for action generation, we scale up training datasets that explicitly require the model to output reasoning and planning steps. Moreover, this approach enhances the interpretability of computer-use agents' behavior, laying a solid foundation for future research.

## C.2. Training Details

For AGUVIS based on the Qwen2-VL backbone, we set the maximum pixels for each image to $1280 \times 720$ to achieve a better trade-off between performance and efficiency[1]. Following the SFT strategy in Wang et al. (2024b), we freeze the ViT parameters during training. For AGUVIS based on the LLaVA-OneVision backbone, we adopt the *anyres* strategy, which splits high-resolution images into multiple patches following (Li et al., 2024a). The maximum sequence length of tokens is set to 8192 for all models. We use Adam optimizer (Loshchilov & Hutter, 2019) for both grounding and planning & reasoning training stages and employ a cosine learning rate scheduler with a warm-up ratio of 3% steps. In the grounding stage, we introduce a grounding packing strategy to enhance training efficiency. We conduct an ablation study using the grounding data of website platform to investigate the strategy effectiveness. We observe that it reduces overall GPU hours from 6 hours to 1 hour. Moreover, this strategy even marginally improve the performance of ScreenSpot website split from 73.3 to 76.8.

We train AGUVIS with a batch size of 128 for 1 epoch in each stage. The peak learning rate is set to 1e-5 for AGUVIS-7B and 5e-6 for AGUVIS-72B. Our codebase is based on Pytorch (Paszke et al., 2019) and Huggingface Transformers (Wolf et al., 2019). During training, we utilize the strategies of DeepSpeed optimization (Rajbhandari et al., 2020), BF16 format and gradient checkpointing to save GPU memory. We train AGUVIS on a cluster of H100-80G GPUs: AGUVIS-7B uses 8 nodes and completes the grounding training within 5 hours and planning & reasoning training within 1 hour. AGUVIS-72B uses 16 nodes and completes the grounding training within 30 hours and planning & reasoning training within 6 hours.

---

[1]During preliminary experiments, we observe that increasing the maximum pixels to $1920 \times 1080$ does not yield significant improvements on ScreenSpot performance.

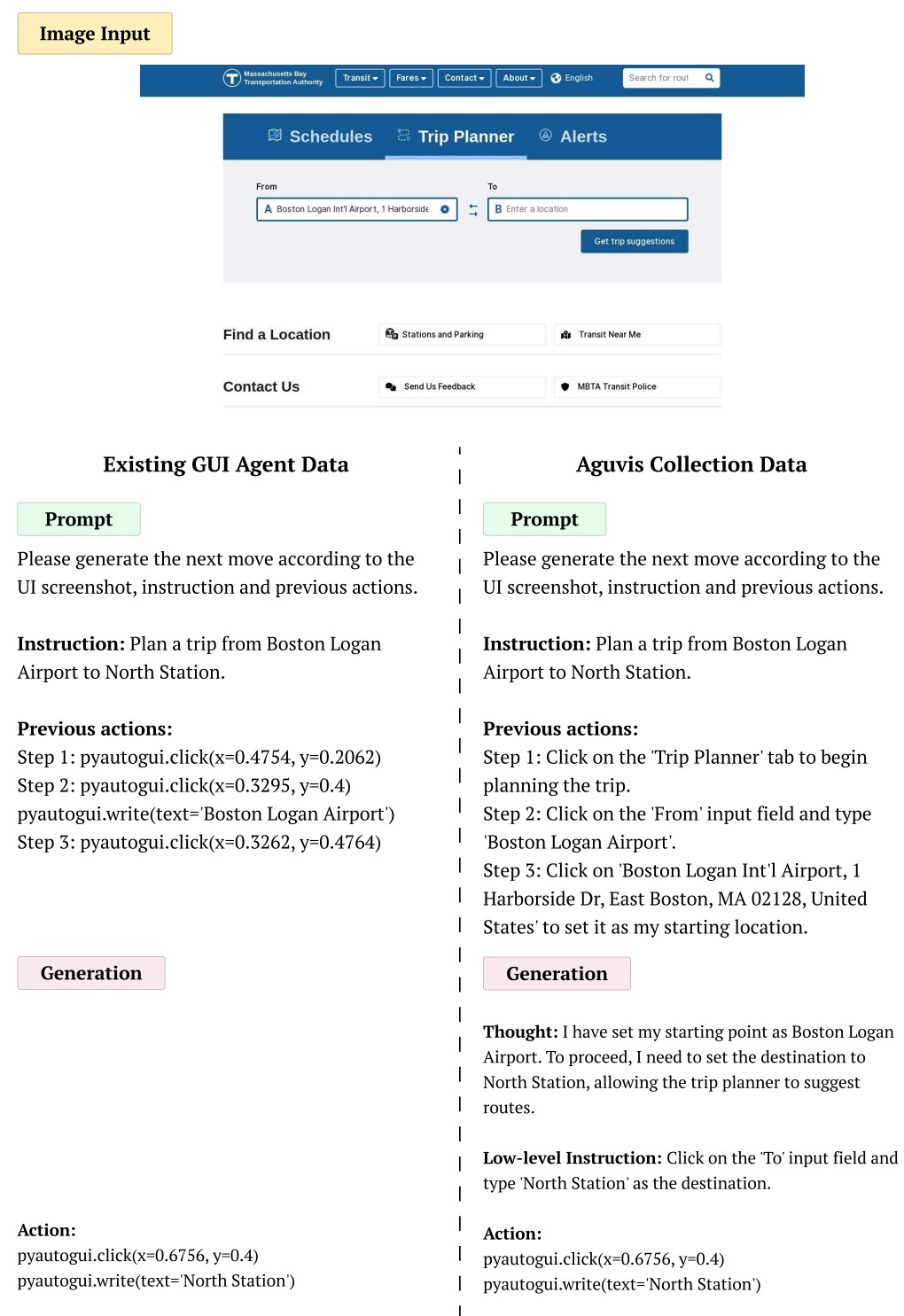

Figure 6: Compared to the schema of exisiting gui agent data (left), the schema of AGUVIS planning & reasoning data (right) includes explicit reasoning process with informative natural language previous action context.

# D. Evaluation Benchmarks

In this section, we introduce more details of evaluation benchmarks used in our work.

## D.1. GUI Grounding Evaluation

**ScreenSpot.** ScreenSpot (Cheng et al., 2024)is a typical benchmark designed specifically for GUI visual grounding, consisting of 1.2K single-step instructions and coordinates of the target elements. This dataset encompasses a variety of grounding instructions tailored for mobile, desktop, and website platforms, and categorizes element types into text and icons/widgets. The benchmark is assessed under two distinct settings: (1) *Original Instructions*: models perform grounding actions directly following the original instructions; and (2) *Self-plan*: models are required to generate plans in natural language based on the original instructions before executing grounding actions.

## D.2. Offline GUI Agent Evaluation

**Multimodal-Mind2Web.** We utilize Multimodal-Mind2Web (Zheng et al., 2024a) for evaluating the offline planning capabilities of GUI agents on websites, which builds on the original Mind2Web (Deng et al., 2023). We report element accuracy (Ele.Acc), Operation F1 (Op.F1), and step success rate (Step SR).

In Table 2 for Multimodal Mind2Web (Zheng et al., 2024a), we only report element accuracy for SeeClick (Cheng et al., 2024) and CogAgent (Hong et al., 2024). This is because the original SeeClick and CogAgent models were evaluated on Mind2Web (Deng et al., 2023), not Multimodal Mind2Web, making the examples misaligned and incomparable. Therefore, we referenced the results from UGround (Gou et al., 2025), where they report the element accuracy of the SeeClick and CogAgent models on Multimodal Mind2Web, striving to comprehensively present all previously representative methods.

**AndroidControl.** Following the setting in Li et al. (2024c), we randomly sample 500 step-actions from AndroidControl full test set to create a subset, and we report the step accuracy on out-of-domain (OOD) data within both high-level and low-level tasks. The high-level task setting necessitates that the model plans and executes actions, whereas the low-level task setting requires the model to simply adhere to human-labeled instructions for executing the next-step action.

## D.3. Online GUI Agent Evaluation

**Mind2Web-Live.** We adopt Mind2Web-Live (Pan et al., 2024) to evaluate GUI agents' online planning, a derived dynamic data set from Mind2Web, comprising 104 real-time interactive web tasks. It evaluates whether each required step within a task has been successfully completed and uses the task success rate (Task SR) as the reported metric. The original Mind2Web-Live is built with WebCanvas (Pan et al., 2024), which is a text-based agent framework. To better accommodate the unified observation and action space of pure vision models, we utilize BrowserGym (Drouin et al., 2024) as the evaluation environment for online web tasks which provide support for pure vision-based agent models. BrowserGym is a browser testing environment built on the Playwright (Microsoft, 2024) engine. We incorporate all Mind2Web-Live tasks and evaluation into BrowserGym, involving registering all Mind2Web-Live tasks, setting up the entry points for these tasks, and porting the Mind2Web-Live evaluation functions to BrowserGym.

As Mind2Web-Live is a text-based benchmark, we have to adapt its evaluation function to suit our pure vision-based model. To achieve this, we introduce the two modifications following:

- For the Mind2Web-Live benchmark's click verification, we adapt our coordinate-based approach by comparing the ground truth CSS selector's bounding box (when available) with our click coordinates, as we cannot directly identify HTML elements.

- Similarly, for input validation, we retrieve and compare the value of the ground truth input element (if present) with the expected value, circumventing the need for precise HTML element identification based on CSS selectors.

The Mind2Web-Live environment relies on real-world websites, many of which implement detection systems for automated browser testing and reCAPTCHA challenges. These factors created difficulties during evluation on the Mind2Web-Live dataset, resulting in a lower task success rate (Task SR). Specifically, we observed the following websites to have significant issues with automation detection:

- **kohls**. Model using the search functionality on the Kohls website through Playwright directly results in a 502 Bad Gateway error.

- **target**. We are unable to open target's job website using Playwright due to network connection error.

- **united**. We are unable to open united website using Playwright due to network connection error.

In addition to the websites that were consistently prone to failure, several other sites intermittently blocked our Playwright access during testing. In total, we encountered 18 network errors and 6 reCAPTCHA tasks that the model was unable to complete, preventing our model from scoring on these 24 tasks.

**AndroidWorld.** AndroidWorld (Rawles et al., 2024b) is a benchmark operating on an Android virtual environment, capable of dynamically instantiating with randomly generated parameters to generate unique tasks for automatic evaluation. It spans 20 real-world applications, encompassing 116 diverse tasks. To assess the pure vision agent models, we follow the instructions in Rawles et al. (2024b), installing a Pixel 6 phone simulator on our computers to serve as the experimental environment. The benchmark incorporates a fully automated task-level evaluation system that automatically assesses whether a state has successfully completed a designated task. The AndroidWorld environment supports optional inputs such as Set-of-Mark (SoM) and textual AXTree information, which most multimodal models currently rely on to complete tasks. However, we solely use raw screenshots as the observation input and restrict the model to coordinate-level actions and basic mobile functions.

**MobileMiniWob.** MobileMiniWob (Rawles et al., 2024b) is the instantiation of 92 tasks from MiniWob++ (Zheng et al., 2024b) in the AndroidWorld environment. Thus, we adopt the same observation and action space used in AndroidWorld and use a real-time evaluation function to determine task success.

### D.3.1. Prompts for using GPT-4o as Planning Model

In all online experiments, we employed two settings: GPT-4o as the planner, AGUVIS-7B as the grounder, and AGUVIS-72B as both the planner and grounder. For experiments where AGUVIS-72B served as both the planner and grounder, the prompt was straightforward: we only needed to provide AGUVIS-72B with a single prompt at each step, and it could independently handle reasoning, planning, and grounding. We use prompt for forcing plan to improve AGUVIS-72B's performance on the online experiments, as illustrated in Appendix E.2.1

In the GPT-4o + AGUVIS-7B setting, the situation was more complex. Two key challenges needed to be addressed: making GPT-4o's planning usable by AGUVIS-7B and determining which actions required AGUVIS-7B for grounding. To address these challenges, we modified GPT-4o's prompts based on Mind2Web-Live (BrowserGym) and AndroidWorld to enable it to delegate grounding actions to AGUVIS-7B when necessary and to share its planning outputs with AGUVIS-7B. Specifically, we append `<|im_start|>assistant<|recipient|>all\nThought:{GPT-4o Thought}\nAction:{GPT-4o Low-level Instruction}` to the end of the prompt and therefore let AGUVIS-7B generate grounding actions based on GPT-4o's response.

Table 12: Prompt used for the planning model in **Mind2Web-Live**, modified from the prompt in (Drouin et al., 2024)

| |
|---|
| **Instructions** 
 Review the current state of the page and all other information to find the best possible next action to accomplish your goal. Your answer will be interpreted and executed by a program, make sure to follow the formatting instructions. |
| **Goal**: {Goal} |
| **Observation of current step** 
 Current URL: {URL} 
 History of interaction with the task: {History} |
| **Action Space** 
 8 different types of actions are available. |

Continued on the next page

Table 12 – Continued from the previous page

**Instructions**

Review the current state of the page and all other information to find the best possible next action to accomplish your goal. Your answer will be interpreted and executed by a program, make sure to follow the formatting instructions.

noop(wait_ms: float = 1000)
Description: Do nothing, and optionally wait for the given time (in milliseconds).

send_msg_to_user(text: str)
Description: Sends a message to the user.

scroll(delta_x: float, delta_y: float, relative: bool = False)
Description: Scroll horizontally and vertically. Amounts in pixels, positive for right or down scrolling, negative for left or up scrolling. Dispatches a wheel event.

fill(element: str, value: str)
Description: Fill out a form field. It focuses the element and triggers an input event with the entered text. It works for <input>, <textarea>, and [contenteditable] elements. The 'element' parameter represents the semantic information of the element you want to fill.

click(element: str, button: Literal['left', 'middle', 'right'] = 'left')
Description: Click an element. The 'element' parameter represents the semantic information of the element you want to click.

dblclick(element: str, button: Literal['left', 'middle', 'right'] = 'left')
Description: Double click an element. The 'element' parameter represents the semantic information of the element you want to double click.

hover(element: str)
Description: Hover over an element. The 'element' parameter represents the semantic information of the element you want to hover over.

keyboard_press(key: str)
Description: Press a combination of keys. Accepts the logical key names that are emitted in the keyboardEvent.key property of the keyboard events: Backquote, Minus, Equal, Backslash, Backspace, Tab, Delete, Escape, ArrowDown, End, Enter, Home, Insert, PageDown, PageUp, ArrowRight, ArrowUp, F1 - F12, Digit0 - Digit9, KeyA - KeyZ, etc. You can alternatively specify a single character you'd like to produce such as "a" or "#". Following modification shortcuts are also supported: Shift, Control, Alt, Meta.

Only a single action can be provided at once. Example:
fill('comment text area', 'This is an example')
Note: you are on mac so you should use Meta instead of Control for Control+C etc.

Table 13: Prompts used for the planning model in **AndroidWorld**, modified from the prompt in (Rawles et al., 2024a)

**Instruction**

You are an agent who can operate an Android phone on behalf of a user. Based on user's goal/request, you may
- Answer back if the request/goal is a question (or a chat message), like user asks "What is my schedule for today?".

*Continued on the next page*

- Complete some tasks described in the requests/goals by performing actions (step by step) on the phone.

When given a user request, you will try to complete it step by step. At each step, you will be given the current screenshot and a history of what you have done (in text). Based on these pieces of information and the goal, you must choose to perform one of the action in the following list (action description followed by the JSON format) by outputing the action in the correct JSON format.
- If you think the task has been completed, finish the task by using the status action with complete as goal_status: {"action_type": "status", "goal_status": "complete"}
- If you think the task is not feasible (including cases like you don't have enough information or can not perform some necessary actions), finish by using the 'status' action with infeasible as goal_status: {"action_type": "status", "goal_status": "infeasible"}
- Answer user's question: {"action_type": "answer", "text": "answer_text"}
- Click/tap on an element on the screen. Please describe the element you want to click using natural language. {"action_type": "click", "target": target_element_description}.
- Long press on an element on the screen, similar with the click action above, use the semantic description to indicate the element you want to long press: {"action_type": "long_press", "target": target_element_description}.
- Type text into a text field (this action contains clicking the text field, typing in the text and pressing the enter, so no need to click on the target field to start), use the semantic description to indicate the target text field: {"action_type": "input_text", "text": text_input, "target": target_element_description}
- Press the Enter key: {"action_type": "keyboard_enter"}
- Navigate to the home screen: {"action_type": "navigate_home"}
- Navigate back: {"action_type": "navigate_back"}
- Scroll the screen or a scrollable UI element in one of the four directions, use the same semantic description as above if you want to scroll a specific UI element, leave it empty when scroll the whole screen: {"action_type": "scroll", "direction": up, down, left, right, "element": optional_target_element_description}
- Open an app (nothing will happen if the app is not installed): {"action_type": "open_app", "app_name": name}
- Wait for the screen to update: {"action_type": "wait"}

**Guidelines**
Here are some useful guidelines you need to follow:
General:
- Usually there will be multiple ways to complete a task, pick the easiest one. Also when something does not work as expected (due to various reasons), sometimes a simple retry can solve the problem, but if it doesn't (you can see that from the history), SWITCH to other solutions.
- Sometimes you may need to navigate the phone to gather information needed to complete the task, for example if user asks "what is my schedule tomorrow", then you may want to open the calendar app (using the 'open_app' action), look up information there, answer user's question (using the 'answer' action) and finish (using the 'status' action with complete as goal_status).
- For requests that are questions (or chat messages), remember to use the 'answer' action to reply to user explicitly before finish! Merely displaying the answer on the screen is NOT sufficient (unless the goal is something like "show me ...").
- If the desired state is already achieved (e.g., enabling Wi-Fi when it's already on), you can just complete the task.
Action Related:
- Use the 'open_app' action whenever you want to open an app (nothing will happen if the app is not installed), do not use the app drawer to open an app unless all other ways have failed.
- Use the 'input_text' action whenever you want to type something (including password) instead of clicking characters on the keyboard one by one. Sometimes there is some default text in the text field you want to type in, remember to delete them before typing.
- For 'click', 'long_press' and 'input_text', the target_element_description parameter you choose must based on a VISIBLE element in the screenshot.

Table 13 – Continued from the previous page

- Consider exploring the screen by using the 'scroll' action with different directions to reveal additional content.
- The direction parameter for the 'scroll' action can be confusing sometimes as it's opposite to swipe, for example, to view content at the bottom, the 'scroll' direction should be set to "down". It has been observed that you have difficulties in choosing the correct direction, so if one does not work, try the opposite as well.
Text Related Operations:
- Normally to select certain text on the screen: (i) Enter text selection mode by long pressing the area where the text is, then some of the words near the long press point will be selected (highlighted with two pointers indicating the range) and usually a text selection bar will also appear with options like 'copy', 'paste', 'select all', etc. (ii) Select the exact text you need. Usually the text selected from the previous step is NOT the one you want, you need to adjust the range by dragging the two pointers. If you want to select all text in the text field, simply click the 'select all' button in the bar.
- At this point, you don't have the ability to drag something around the screen, so in general you can not select arbitrary text.
- To delete some text: the most traditional way is to place the cursor at the right place and use the backspace button in the keyboard to delete the characters one by one (can long press the backspace to accelerate if there are many to delete). Another approach is to first select the text you want to delete, then click the backspace button in the keyboard.
- To copy some text: first select the exact text you want to copy, which usually also brings up the text selection bar, then click the 'copy' button in bar.
- To paste text into a text box, first long press the text box, then usually the text selection bar will appear with a 'paste' button in it.
- When typing into a text field, sometimes an auto-complete dropdown list will appear. This usually indicating this is a enum field and you should try to select the best match by clicking the corresponding one in the list.

# E. Analysis

## E.1. More Training Ablation

### E.1.1. TRAINING STRATEGY ABLATION

To further demonstrate the contribution of Stage 1 (GUI Grounding), Stage 2 (GUI Planning & Reasoning), and their combination to model training, we conducted an ablation study. Specifically, we designed five experimental settings on AGUVIS$_{\text{QWEN2-VL}}$ and AGUVIS$_{\text{LLAVA-OV}}$:

- **Stage 1 → Stage 2** corresponds to the staged configuration AGUVIS used in our paper, where Stage 1 is followed by Stage 2 sequentially.

- **Stage 1 + Stage 2** represents a joint training setup, where two stages are combined into a training process.

- **w/o Stage x** indicates the absence of the respective stage in the setting.

Note that for the setting of removing Stage 2 (w/o Stage 2 or w/o Stage 1 & 2), the models are fine-tuned on the corresponding task-specific dataset for planning tasks.

From the first two rows in Table 14, it can be observed that the differences between models trained with Staged Training and Joint Training setups are relatively minor. However, a clear trend emerges: models trained using the Joint Training setup perform better on GUI grounding tasks but exhibit inferior performance on datasets requires planning ability such as MM-Mind2Web and AndroidControl High-level. This trend implies grounding data in Stage 1 is more abundant, dominating the optimization process and biasing the model toward grounding tasks. In contrast, the data in Stage 2, which combines planning and grounding, is of higher quality and better aligned with the agent's deployment scenarios. This rationale underpins our decision to position Stage 2 later in the training sequence.

Moreover, it is observed that compared to AGUVIS$_{\text{QWEN2-VL}}$ trained through both Stage 1 and Stage 2, the model trained with only Stage 2 data maintains similar performance on MM-Mind2Web and AndroidControl but exhibits a notable decline

in GUI grounding performance on ScreenSpot. This suggests that the stability on Mind2Web and AndroidControl can be attributed to Qwen2VL's pre-training on natural image grounding. However, the diverse image and domain requirements of the ScreenSpot GUI grounding test set highlight the necessity of extensive and varied grounding training from Stage 1. This training is essential for improving the grounding performance required for a cross-platform GUI agent model.

To verify this analysis, we conduct the same ablation study on the LLaVA model, as shown in Table 15. From the results, we can see that the original LLaVA did not undergo extensive natural image grounding training during the training process, making it insufficient for LLaVA to excel when only Stage 1 or Stage 2 is conducted. When both Stage 1 and Stage 2 are performed, LLaVA can be significantly improved, even surpassing previous SOTA results. This validates the above analysis and further demonstrates that our method is model-agnostic and universally applicable to popular VLMs like Qwen2-VL and LLaVA.

Table 14: Ablation study of AGUVIS$_{\text{QWEN2-VL}}$ on training strategy.

| Settings | ScreenSpot | Multimodal-Mind2Web | | | AndroidControl | |
|---|---|---|---|---|---|---|
| | | Cross-Task | Cross-Website | Cross-Domain | High-Level | Low-Level |
| Stage 1 → 2 | 84.4 | 58.5 | 55.4 | 54.8 | 61.5 | 80.5 |
| Stage 1 + 2 | 85.0 | 56.1 | 53.1 | 55.6 | 59.2 | 80.9 |
| w/o Stage 2 | 81.8 | 50.9 | 45.2 | 45.3 | 58.0 | 75.6 |
| w/o Stage 1 | 77.4 | 59.7 | 55.3 | 55.8 | 58.8 | 79.8 |
| w/o Stage 1 & 2 | 55.3 | 50.9 | 44.9 | 47.7 | 59.1 | 59.2 |

Table 15: Ablation study of AGUVIS$_{\text{LLAVA-OV}}$ on training strategy.

| Settings | ScreenSpot | Multimodal-Mind2Web | | | AndroidControl | |
|---|---|---|---|---|---|---|
| | | Cross-Task | Cross-Website | Cross-Domain | High-Level | Low-Level |
| Stage 1 → 2 | 81.2 | 55.3 | 50.0 | 50.8 | 60.7 | 82.4 |
| w/o Stage 2 | 70.0 | 43.4 | 39.0 | 40.7 | 54.9 | 65.6 |
| w/o Stage 1 | 71.3 | 42.5 | 40.3 | 42.8 | 61.4 | 80.5 |
| w/o Stage 1 & 2 | 3.8 | 33.8 | 30.5 | 32.4 | 50.4 | 50.0 |

### E.1.2. DATA STRATEGY ABLATION

To investigate the impact of different device domain datasets within a unified action space, we designed three settings on the MM-Mind2Web dataset: (1) training with the complete dataset comprising both Web and Mobile data, (2) training using only the Web data, and (3) fine-tuning exclusively on the MM-Mind2Web dataset. All three experiments include fine-tuning on the MM-Mind2Web dataset.

Table 16: Ablation Study of The Impact of Mobile Data on MM-Mind2Web.

| Model | Training Data | MM-Mind2Web | | |
|---|---|---|---|---|
| | | Cross-Task | Cross-Website | Cross-Domain |
| AGUVIS$_{\text{QWEN2-VL}}$ | Web + Mobile (Stage 2 Equivalent) | 58.5 | 55.4 | 54.8 |
| | Web Only | 53.1 | 50.3 | 52.2 |
| | Mind2Web Only | 50.9 | 44.9 | 47.7 |
| AGUVIS$_{\text{LLAVA-OV}}$ | Web + Mobile (Stage 2 Equivalent) | 55.3 | 50.0 | 50.8 |
| | Web Only | 44.9 | 43.5 | 42.1 |
| | Mind2Web Only | 43.4 | 39.0 | 40.7 |

Table 17: Ablation Study of the Impact of Inner Monologue.

| AGUVIS | ScreenSpot | Multimodal-Mind2Web | | | AndroidControl | |
|---|---|---|---|---|---|---|
| | | Cross-Task | Cross-Website | Cross-Domain | High-Level | Low-Level |
| AGUVIS | 84.4 | 58.5 | 55.4 | 54.8 | 61.5 | 80.5 |
| AGUVIS w/o IM | 79.3 | 55.4 | 53.7 | 54.9 | 60.3 | 69.1 |

The experimental results, presented in the Table 7, demonstrate that training AGUVIS with both Web and Mobile data consistently outperforms the setting trained exclusively on MM-Mind2Web. This performance gain underscores the contribution of Mobile data to enhancing cross-device domain generalization in the Web domain, validating the effectiveness of our cross-platform data.

In addition, we conducted ablation study on the role of incorporating inner monologue (IM) in training. The result shown in Table 17 demonstrated clear performance gain from inner monologue. This gain can be attributed to two key factors: the use of inner monologue enables the model to elicit reasoning about the current step while also serving as context to facilitate more effective planning for subsequent steps. Additionally, incorporating low-level instructions from the training data improves the accuracy of the model's action execution, as demonstrated in both the Screenspot and AndroidControl low-level tasks. This is mainly because the low-level instructions of inner monologue act as atomic instruction and grounding action pairs, also enhancing the grounding ability of our GUI agents.

### E.2. Planning Analysis

#### E.2.1. PROMPTS FOR SELF-PLANNING AND ENFORCED PLANNING MODE.

In Appendix C.1, we present the training data schema for Stage 1 and Stage 2. We use the special token `<|recipient|>` along with `os` or `all` to control whether the message content is an inner monologue or a pyautogui action command. Thanks to this design, we can use `<|recipient|>` during the inference phase to control the content generated by the agent model.

In the *Enforced Plan* Setting, we employ the `<|recipient|>all\nThought` prompt to compel the model to generate a planning phase following this. This setting explicitly requires the model to utilize inner monologue for high-level reasoning before taking actions. While in the *Self-plan* setting, we do not add any word after `<|recipient|>`, so the model can choose to generate `os` to directly produce a `pyautogui` command, or generate `all` to first create natural language reasoning and then generate a `pyautogui` command. Thus, the model can autonomously determine whether to generate planning thoughts based on the complexity of tasks.

As noted in Section 4.5, the enforced planning resolves approximately 20% of grounding errors by encouraging the model to carefully consider the task context, potential ambiguities, and available UI elements before committing to action.

---

**Prompt Template For Self-plan Setting**

```
<|im_start|>system
You are a GUI agent. You are given a task and a screenshot of the screen. You need
to perform a series of pyautogui actions to complete the task.<|im_end|>
<|im_start|>user
<|vision_start|><|image_pad|><|vision_end|>Please generate the next move according
to the ui screenshot, instruction and previous actions.

Instruction: {goal}

Previous actions: {previous_actions}
<|im_end|>
<|im_start|>assistant<|recipient|>
```

Prompt Template For Enforced Plan Setting

```
<|im_start|>system
You are a GUI agent. You are given a task and a screenshot of the screen. You need
to perform a series of pyautogui actions to complete the task.<|im_end|>
<|im_start|>user
<|vision_start|><|image_pad|><|vision_end|>Please generate the next move according
to the ui screenshot, instruction and previous actions.

Instruction: {overall_goal}

Previous actions: {previous_actions}
<|im_end|>
<|im_start|>assistant<|recipient|>all
Thought:
```

E.2.2. CASES OF INNER MONOLOGUE BONUS

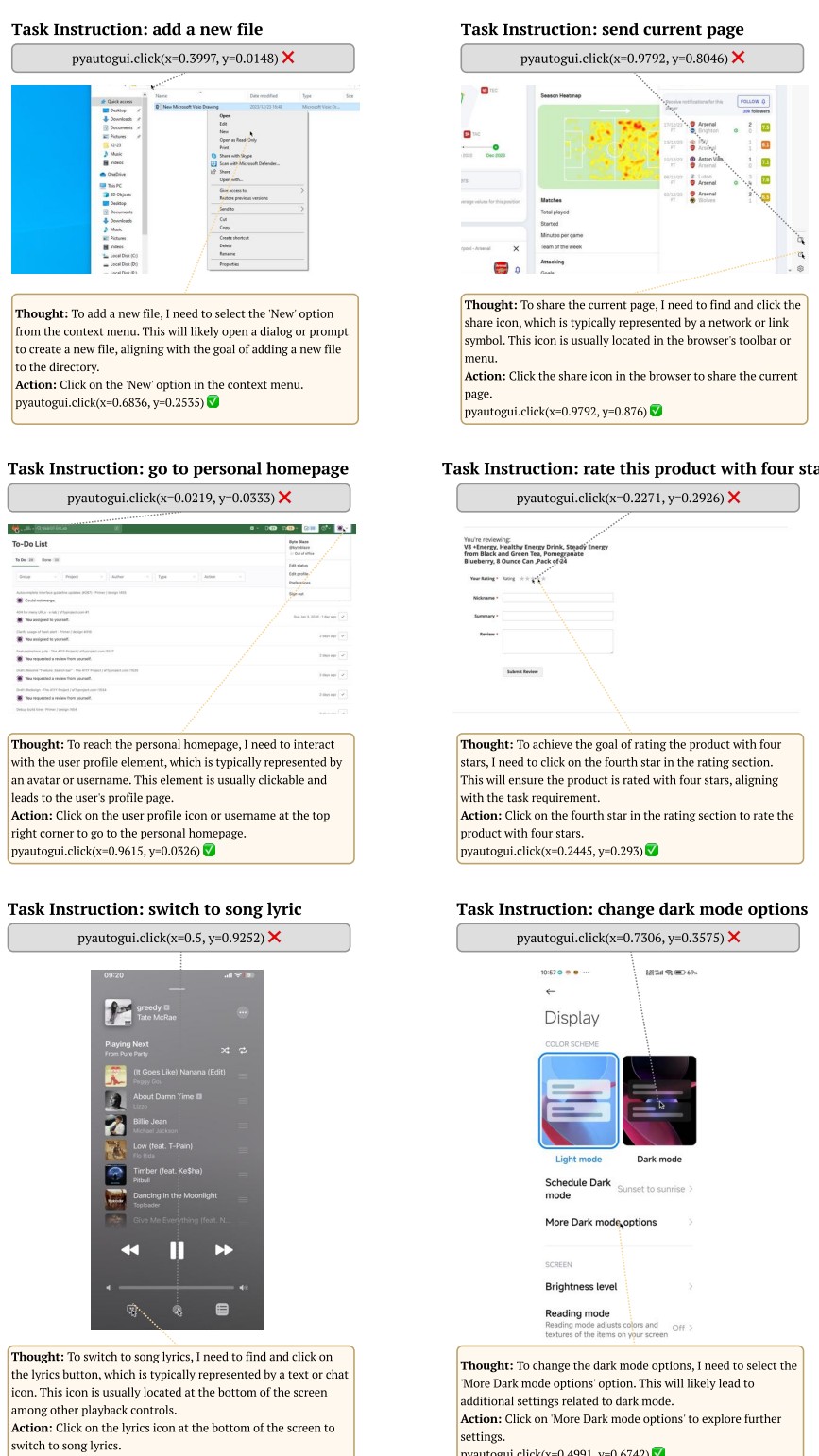

Figure 7: Benefits of inner monologue in autonomous GUI interactions across Desktop, Website, and Mobile environments. Gray indicates output without inner monologue, yellow with inner monologue.

## E.3. AGUVIS Trajectories Examples on Online Evaluation

### E.3.1. MIND2WEB-LIVE CASE: AGUVIS-72B AS PLANNER AND GROUNDER

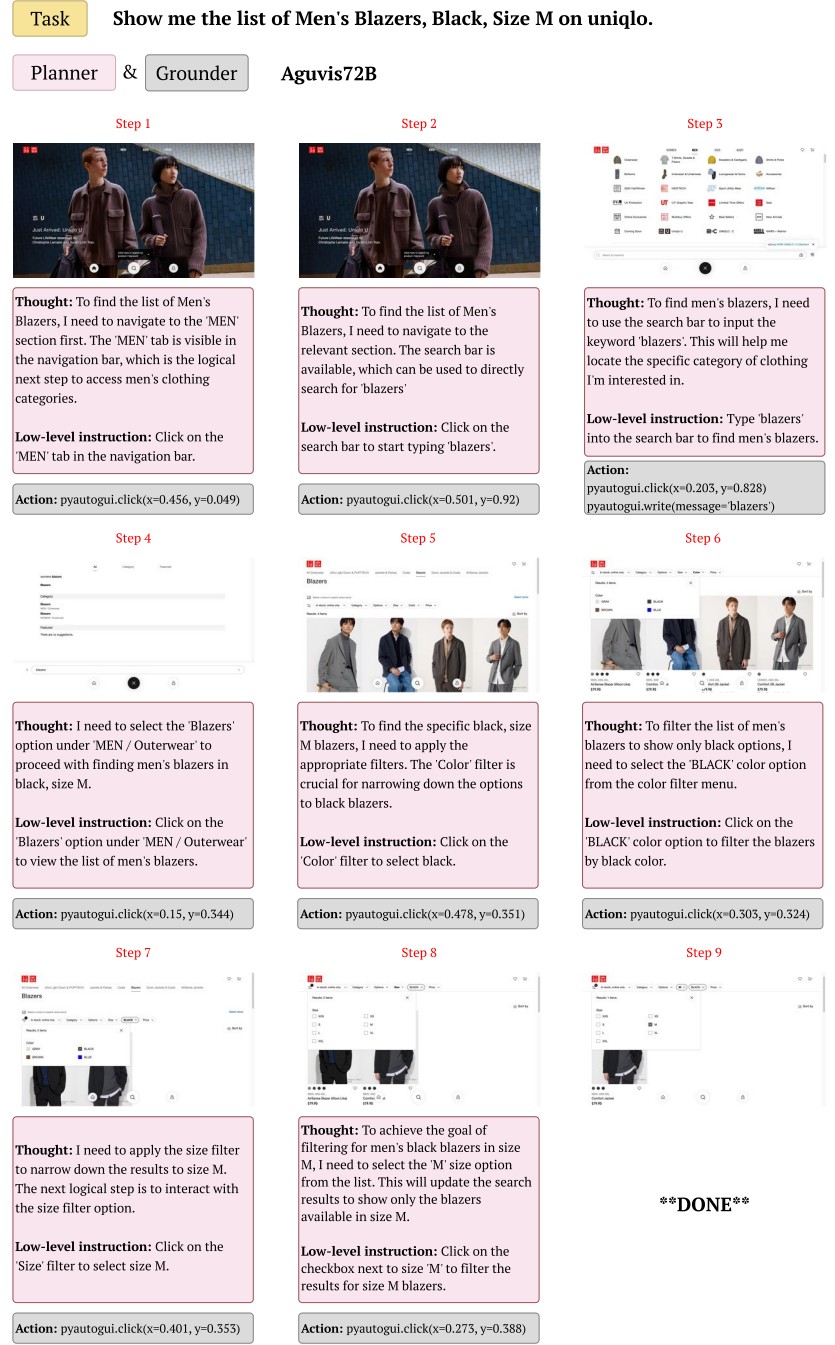

Figure 8: Example of AGUVIS-72B as planner and grounder executing on Mind2Web-Live task. Due to space limitations, we present here the trajectory generated guided by `Thought`.

### E.3.2. Mind2Web-Live Case: GPT-4o as Planner and Aguvis-7B as Grounder

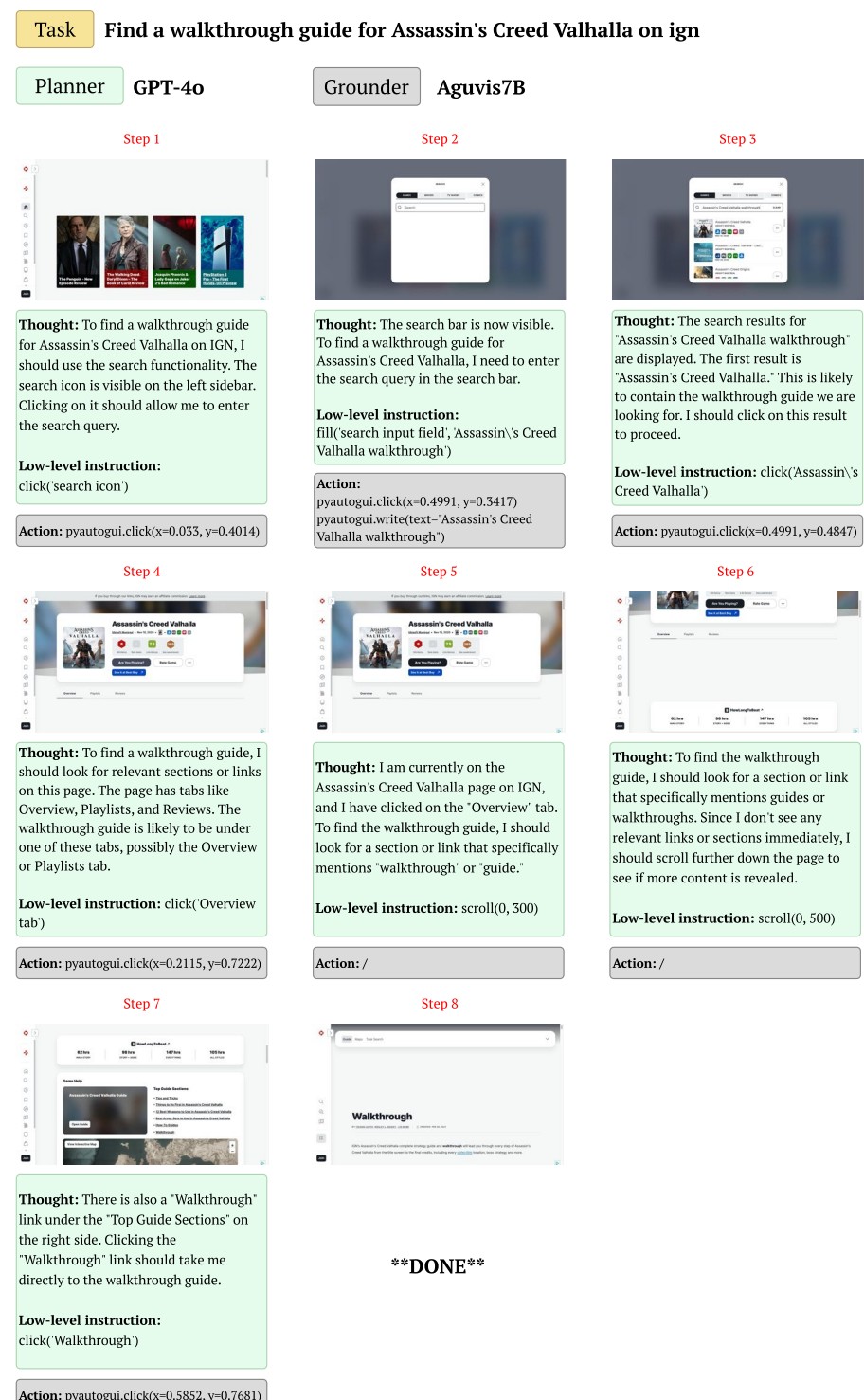

Figure 9: Example of GPT-4o as planner and AGUVIS-7B as grounder executing on Mind2Web-Live task.

## E.3.3. ANDROIDWORLD CASE: AGUVIS-72B AS PLANNER AND GROUNDER

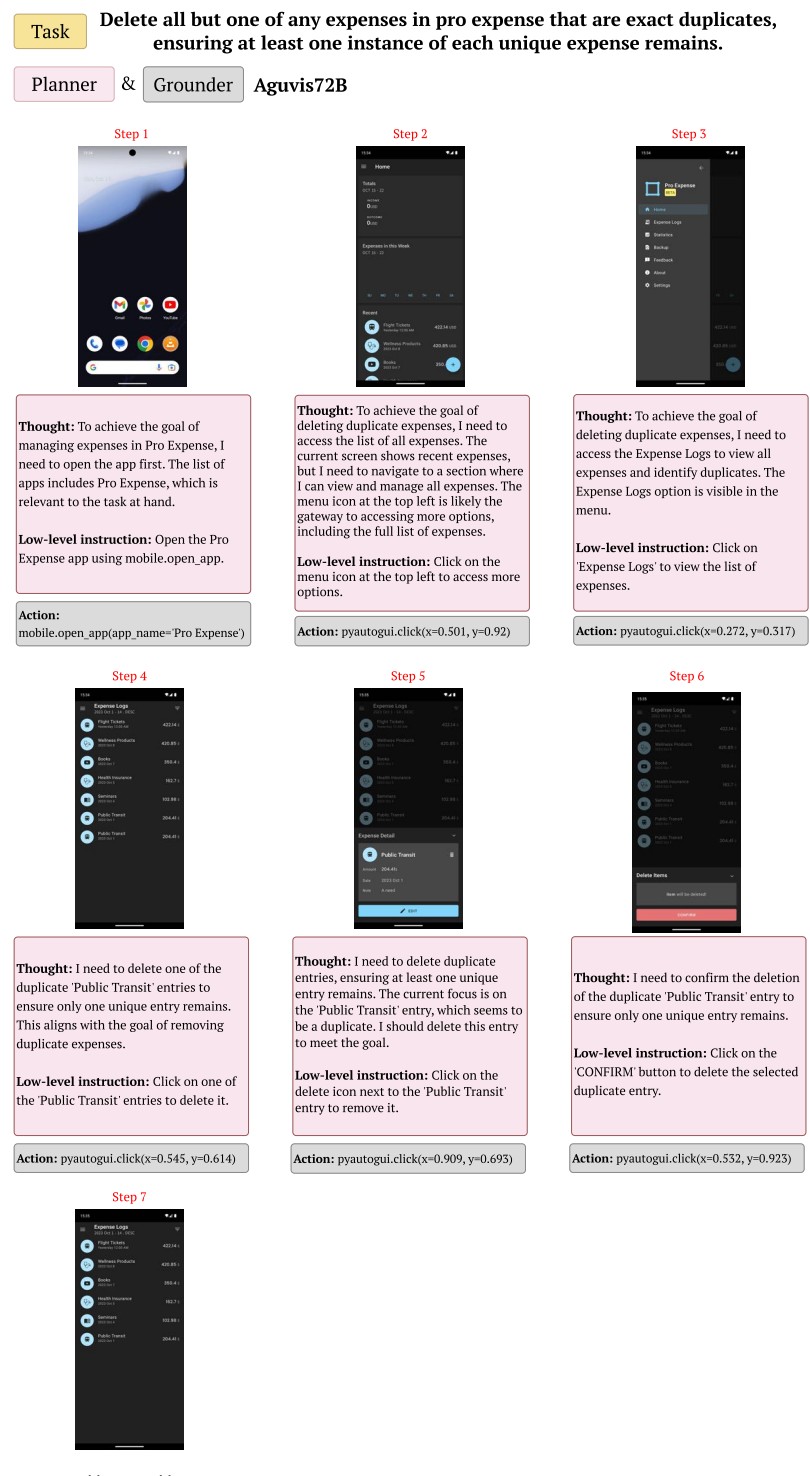

Figure 10: Example of AGUVIS-72B as planner and grounder executing on AndroidWorld task. Due to space limitations, we present here the trajectory generated guided by `Thought`.

E.3.4. ANDROIDWORLD CASE: GPT-4O AS PLANNER AND AGUVIS-7B AS GROUNDER

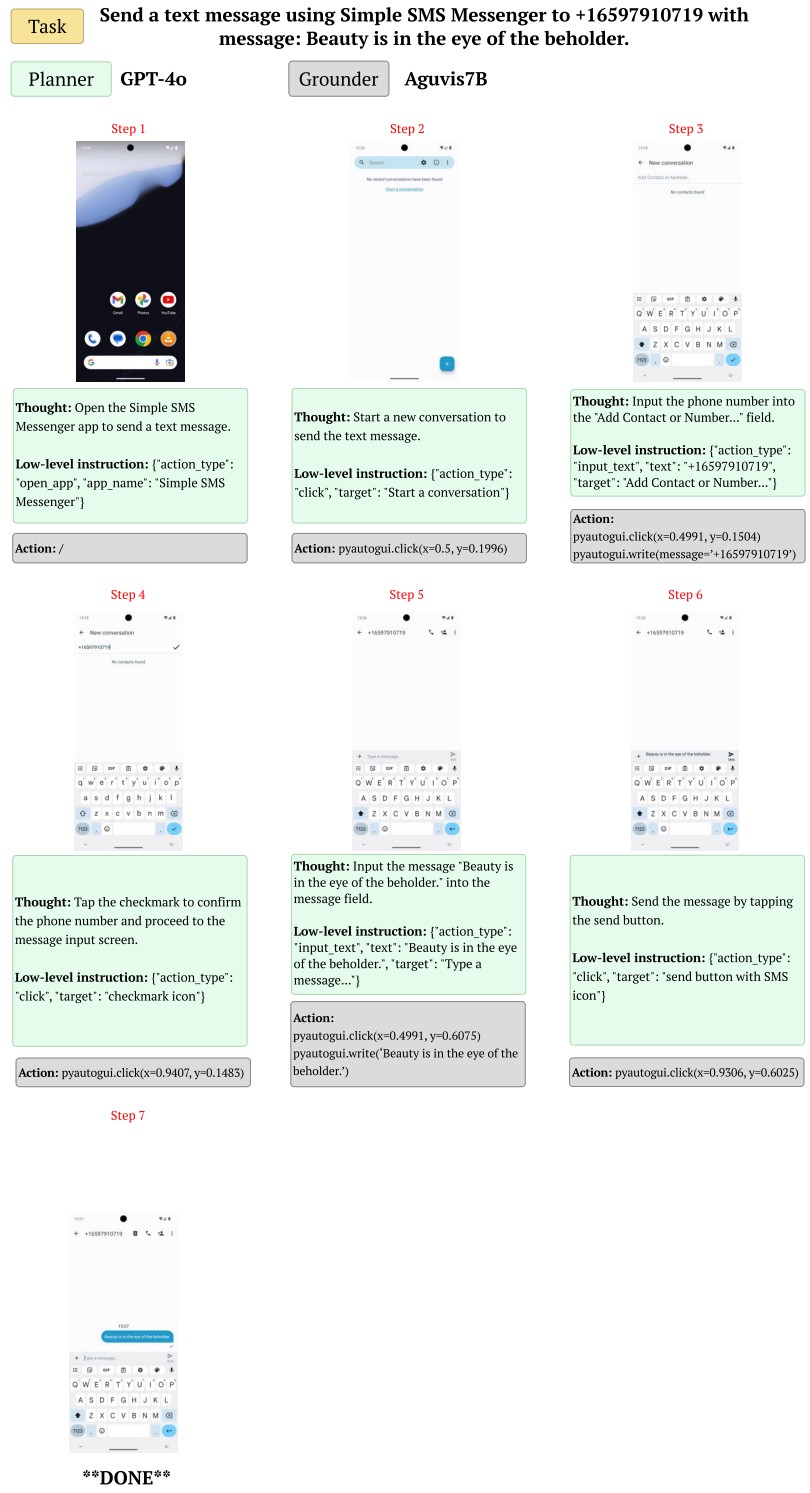

Figure 11: Example of GPT-4o as planner and AGUVIS-7B as grounder executing on AndroidWorld task.

**E.4. Case of AGUVIS Generalization in Real-World Scenarios**

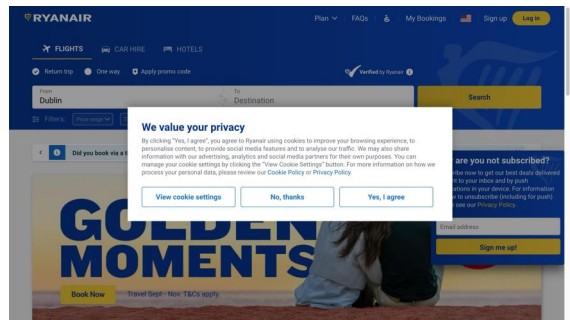 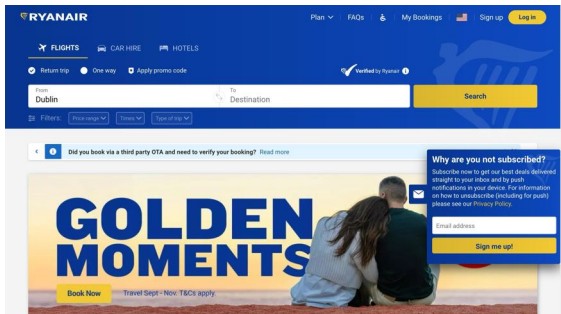

**Thought:** To proceed with finding a flight, I need to dismiss the privacy pop-up to access the search functionality.
**Low-level instruction:** Click 'No, thanks' to dismiss the privacy pop-up.
**Action:** pyautogui.click(x=0.498, y=0.603)

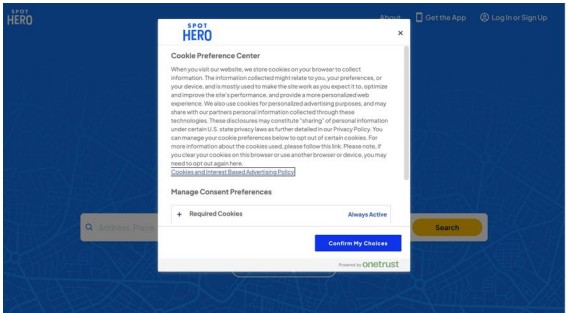 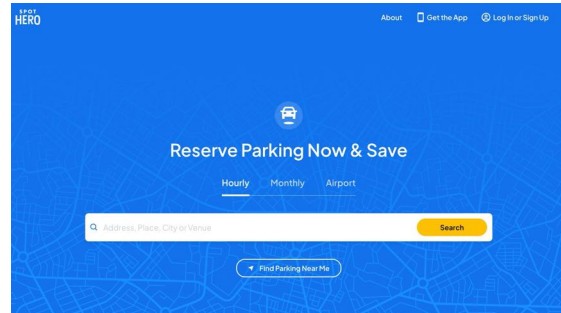

**Thought:** To proceed with contacting support, I need to close this overlay to access the main page where the 'Contact Us' or 'Support' link is likely located.
**Low-level instruction:** Click the 'X' button to close the cookie consent overlay.
**Action:** pyautogui.click(x=0.704, y=0.093)

Figure 12: Case of AGUVIS generalization in real-world scenarios: closing cookie pop-ups, which is an out-of-domain situation in our training data.

