# OpenReview forum: "Aguvis: Unified Pure Vision Agents for Autonomous GUI Interaction"
_ICML.cc/2025/Conference — ICML 2025 poster_

### Official Review · Reviewer_zR1B · 2025-03-13

**Overall Recommendation:** 4

**Summary:**

This paper focuses on the autonomous GUI interaction task from the pure vision agent perspective. A large-scale cross-platform dataset of GUI agent trajectories is constructed. A two-stage training pipeline is proposed to separate GUI grounding from planning and reasoning. The experiments demonstrate the effectiveness of the proposed method in both offline and real-world online benchmarks.

## update after rebuttal

I appreciate the authors' clarifications. Most of my concerns have been addressed by the rebuttal. I would lean to accept the paper by involving the additional discussions in the revised version.

**Claims And Evidence:**

Yes.

**Essential References Not Discussed:**

No.

**Experimental Designs Or Analyses:**

The current experiments are not thorough enough. Specifically, the evaluation metrics on different benchmarks are not the same. For example, the computational cost analysis is only conducted on Mind2Web-Live, and the element accuracy and Operation F1 are only conducted on Multimodal Mind2Web. To make the results more convincing, it would be better to add additional evaluations and discussions.

**Methods And Evaluation Criteria:**

Yes.

**Other Comments Or Suggestions:**

The first paragraph in Section 3.3 claims that “we evaluated AGUVIS across four comprehensive benchmarks”. But only three benchmarks are mentioned here.

**Other Strengths And Weaknesses:**

The proposed two-stage training pipeline could incorporate structured thought processes to enhance the performance of autonomous vision-based GUI agents. The collected dataset could also be useful for future research in the community.

**Questions For Authors:**

1. Which metric is used for the result comparison in Table 1?

2. Why are the evaluation metrics used on different benchmark datasets different? For example, the computational cost analysis is only conducted on Mind2Web-Live, and the element accuracy and Operation F1 are only conducted on Multimodal Mind2Web.

3. Based on the results in Table 8, it seems that the proposed method paired with GPT-4o for planning, the result is better than that of the AGUVIS-72B. I was wondering about the results of the same configuration on other benchmarks.

4. For the VLM-based trajectory augmentation process, the current inner monologue components are generated by GPT-4o. Can other latest VLMs be used and compared?

**Relation To Broader Scientific Literature:**

The proposed datasets and models could benefit future research in autonomous vision-based GUI agents.

**Theoretical Claims:**

Yes.

---

> ### Author Rebuttal · Authors · 2025-04-01
>
> Thank you for taking the time to review our work and providing constructive feedback! We greatly appreciate your recognition of our contribution in the realm of pure-vision based autonomous GUI agents, including our grounding-then-planning training pipeline and open-sourcing the large-scale, curated data. We are also delighted that you recognized the promising performance of our approach in both offline and real-world online benchmarks, which highlights its generalizability across multiple digital platforms. We have fully open-sourced our roadmap towards building pure-vision based autonomous agents, and are committed to support further research in this field.
>
> We also noticed you have some constructive questions about our work, and we're happy to elaborate further below!
>
> > C1: The first paragraph in Section 3.3 claims that “we evaluated AGUVIS across four comprehensive benchmarks”. But only three benchmarks are mentioned here.
>
> Thank you for pointing out this inconsistency! We will correct it in the revised version of the manuscript.
>
> > Q1: Which metric is used for the result comparison in Table 1?
>
> For Table 1 (results on Screenspot), we report Click Accuracy as the evaluation metric, following the definitions provided in the original benchmark paper [1]. Specifically, Click Accuracy measures the proportion of test samples where the predicted location falls within the ground truth element's bounding box.
>
> [1] SeeClick: Harnessing GUI Grounding for Advanced Visual GUI Agents. Cheng et al., 2024.
>
> > Q2: Why are the evaluation metrics used on different benchmark datasets different? For example, the computational cost analysis is only conducted on Mind2Web-Live, and the element accuracy and Operation F1 are only conducted on Multimodal Mind2Web.
>
>
> Thank you for your thoughtful feedback! The differences in evaluation metrics arise because each benchmark is designed to assess different aspects of the agent's performance, and we adhere to the metric definitions specific to each benchmark. This approach ensures that our results remain comparable to previous methods and provides a clear understanding of the agent's performance under distinct conditions.
>
> - **Offline vs. Online Evaluations**: Some metrics, such as element accuracy and Operation F1, are more suitable for offline evaluations (e.g., Multimodal Mind2Web), where the goal is to evaluate step-level accuracy and action predictions without environments. In contrast, online evaluations like those on Mind2Web-Live focus on final trajectory-level execution results, which offer a more holistic view of the agent's ability to complete tasks in dynamic environments.
>
> - **Benchmark-Specific Metrics**: Different benchmarks introduce their own unique metrics to capture specific aspects of performance. For instance, Mind2Web-Live introduces the USD Efficiency Score, which evaluates the efficiency of resource utilization during task execution. This metric provides insights into the agent’s performance in real-world settings, where efficiency is a key factor.
>
> While we acknowledge the importance of following standardized evaluation metrics, we also agree that it is valuable to align the evaluation criteria with the specific needs and goals of each benchmark. Developing unified metrics across benchmarks requires significant effort but remains an important direction for our future work.
>
> > Q3: Based on the results in Table 8, it seems that the proposed method paired with GPT-4o for planning, the result is better than that of the AGUVIS-72B. I was wondering about the results of the same configuration on other benchmarks.
>
> We appreciate your interest in this observation! In addition to the AGUVIS-7B paired with GPT-4o for planning shown in Table 8 of OSWorld, we have also applied this configuration to other online GUI agent benchmarks, such as Mind2Web-Live, AndroidWorld, and MobileMiniWob, as detailed in Tables 4 and 5.
>
>
> > Q4: For the VLM-based trajectory augmentation process, the current inner monologue components are generated by GPT-4o. Can other latest VLMs be used and compared?
>
> Our data processing pipeline is model-agnostic and can be extended to more recent VLMs. Our data pipeline utilizes a VLM annotator to generate inner monologue reasoning from human-annotated actions. This is a natural fit for human behavior, where humans easily generate actions but find it costly to record their inner monologue. Conversely, current VLM struggle to make accurate action decisions but can more easily infer inner monologues given action decisions. This novel approach provides a promising pipeline for building agent trajectories with reasoning. We believe such annotations could equivalently be generated by open-source VLMs. Although time and cost constraints during rebuttal have prevented us from fully re-collecting data and retraining with alternative models, we believe that exploring other open-source VLMs for this process is an exciting avenue for future work!

---

### Official Review · Reviewer_X3pb · 2025-03-14

**Overall Recommendation:** 3

**Summary:**

The paper introduces AGUVIS, a unified vision-based framework for autonomous GUI agents designed to overcome limitations of existing approaches, which rely on textual representations, platform-specific actions, and closed-source models for reasoning. AGUVIS enables direct operation on screen images, standardizes cross-platform interactions via a plugin system, and incorporates inner monologue-structured reasoning through explicit thought processes—to handle complex tasks requiring planning.

**Claims And Evidence:**

They are clear.

**Essential References Not Discussed:**

N/A

**Experimental Designs Or Analyses:**

Yes

**Methods And Evaluation Criteria:**

Yes.

**Other Comments Or Suggestions:**

N/A

**Other Strengths And Weaknesses:**

Strengths:
1. It achieves good results across offline and real-world online benchmarks.
2. It provides an effective framework to collect the GUI grounding and planning data.

Weaknesses:
1. The training method proposed in this paper does not have technical innovation. Most of the existing end-to-end GUI Agent training methods include Grounding pre-training and decision training[1,3,4], which is a general idea.
2. I do not agree about what authors claim in.the paper “first fully autonomous vision-based GUI agent that operates without relying on closed-source models”. Firstly, it is disputable that data collection pipline in this paper also need to leverage closed-source model (GPT). This is essentially distilling the common sense capability in from closed source models. Additionally, several studies have attempted to create algorithms that can enable open-source models to achieve the same or even better performance compared to the current open-source models [1,2,3]. This statement exaggerates the contribution of the article.
3. During the testing, is the model tested based on the setting of multi-image trajectories, or on the setting of text history + single image? Suffice it to say, is a setting based on multi-graph trajectories more appropriate.
4. I suggest author to consider to evaluate the model on other frequently-used datasets (e.g., AITZ[4]).

[1]. OS-ATLAS: A Foundation Action Model for Generalist GUI Agents
[2]. InfiGUIAgent: A Multimodal Generalist GUI Agent with Native Reasoning and Reflection
[3]. GUI Odyssey: A Comprehensive Dataset for Cross-App GUI Navigation on Mobile Devices
[4]. MobileVLM: A Vision-Language Model for Better Intra- and Inter-UI Understanding
[5]. Android in the Zoo: Chain-of-Action-Thought for GUI Agents

**Questions For Authors:**

When comparing GPT-4o+AGUVIS-7B and AGUVIS-72B, I found that the conclusion on Mind2Web-Live （Table 4) is not consistent with the conclusion on AndroidWorld (Table 5). Can you explain the reason?

**Relation To Broader Scientific Literature:**

Related to the GUI understanding and Agent Reasoning.

**Theoretical Claims:**

Proofs for theoretical claims are correct

---

> ### Author Rebuttal · Authors · 2025-04-01
>
> We sincerely appreciate your thoughtful review and the opportunity to further clarify our contributions.
>
> > **W1: Technical Innovation in Training Method**
>
> Thank you for your insightful comments regarding the training methodology! We recognize that recent approaches share similar high-level components, such as grounding pre-training and decision training. However, AGUVIS's primary innovation lies in its unified framework that incorporates inner monologue reasoning. This enables the agent to operate effectively in new, previously unseen platforms without additional retraining. This integration, along with our training methods, distinctly sets AGUVIS apart from recent methods [1,3,4].
>
> These innovations are empirically validated:
>
> - Section 4.1 shows that our two-stage training strategy outperforms joint training and ablations.
> - Section 4.2 highlights how inner monologue enables explicit planning and task decomposition, surpassing reactive action decision methods.
> - Section 4.3 demonstrates cross-platform generalization, where AGUVIS trained on web/mobile tasks transfers well to desktop GUIs in OSWorld.
>
> We would also greatly appreciate it if you could review these contributions highlighted by other reviewers as well. We strongly believe that the framework design, analyses, open-source model, and data contributions could significantly benefit and advance the GUI agent community.
>
> > **W2: Claim of “first fully autonomous vision-based GUI agent that operates without relying on closed-source models”**
>
> Thank you for highlighting this important aspect of our work. Our claim specifically emphasizes that AGUVIS, once trained, operates fully autonomously during task execution without dependence on closed-source models. Crucially, the model itself—including all training data, architectures, and procedures—is completely open-sourced, enabling full transparency and reproducibility.
>
> While our data pipeline leveraged GPT-4o to generate inner monologue reasoning from human-annotated actions, this step was employed solely to enrich reasoning data rather than define action policies. And we believe such annotations could equivalently be generated by open-source VLMs, and exploring this capability with open-source models is definitely an important part of our future work.
>
> Moreover, while recent research ([1, 2, 3]) has similarly aimed at competitive performance using open-source models, AGUVIS uniquely provides a unified vision-based  framework capable of seamlessly operating across multiple diverse GUI environments. We will further clarify this comparative context in our revised manuscript to better highlight AGUVIS’s contributions and clearly differentiate it from concurrent work.
>
> > **W3. Testing settings with multi-image trajectories vs. text history + single image.**
>
> We appreciate the suggestion. AGUVIS currently uses text history + single image, balancing context richness and computational feasibility. Incorporating multiple images poses significant token cost challenges (~1200 tokens/image), especially when paired with inner monologue reasoning and 72B model size.
>
> Nonetheless, we recognize the potential of multi-image context and plan to explore it in future iterations. Our open-source format supports multi-image inputs, enabling future work to build on this. As models like Qwen2.5-VL with advanced video ability, we anticipate AGUVIS will scale to multi-frame settings more efficiently.
>
> > **W4. Evaluation on other datasets, such as AITZ.**
>
> Thank you for this suggestion. We evaluated AGUVIS on the AITZ benchmark. Results are summarized below and demonstrate AGUVIS’s advanced performance:
>
>
> |Model|Total Match|
> |----|----|
> |CogAgent(Zero-shot)|44.5|
> |CogAgent(CoAT-finetuned) | 53.3 |
> |AUTO-UI(Zero-shot)|34.5|
> |AUTO-UI (CoAT-finetuned)|47.7|
> |OS-Atlas-Pro-7B (CoAT-finetuned)|58.3|
> |AGUVIS-7B|63.3|
> |AGUVIS-72B|66.1|
>
> These results will also be included in our revision to strengthen the paper’s empirical validation.
>
> > **Q1. Discrepancy between results on Mind2Web-Live and AndroidWorld.**
>
> We appreciate your attention to the detailed results. We think the discrepancy between Mind2Web-Live (Table 4) and AndroidWorld (Table 5) may stem from the differences in how GPT-4o understands and interacts with web interfaces versus mobile interfaces. We observed that GPT-4o tends to be distracted by extraneous details in high-resolution, information-rich web interfaces, which can lead to failures in planning. We will clarify these environmental factors and their impact on agent performance in our revision.

---

### Official Review · Reviewer_VD9H · 2025-03-14

**Overall Recommendation:** 3

**Summary:**

This paper introduces Aguvis, a vision-based framework that operates directly on screen images, providing a standardized cross-platform interaction method enhanced by structured reasoning through inner monologue. The researchers developed a comprehensive dataset with multimodal annotations and implemented a two-stage training approach that separately handles GUI grounding and planning. Experimental results demonstrate that Aguvis achieves leading performance on both offline and real-world benchmarks.

**Claims And Evidence:**

The claim is clear, focusing primarily on the data construction methodology with inner monologue in GUI data. The experiments also confirm its effectiveness.

**Essential References Not Discussed:**

The paper has a comprehensive literature review covering most relevant work.

**Experimental Designs Or Analyses:**

The authors employ a comprehensive evaluation across both offline benchmarks (ScreenSpot, Multimodal-Mind2Web, AndroidControl) and online benchmarks (Mind2Web-Live, AndroidWorld, MobileMiniWob).

The ablation studies in Section 4 are well-structured, particularly those examining the impact of training stages, inner monologue, and cross-platform benefits. The error analysis in Section 4.5 provides a balanced view of the model's limitations.

**Methods And Evaluation Criteria:**

The mode architecture's innovation is relatively weak, as it appears quite similar to the baseline Qwen2-VL. However, the paper mainly proposes a new pipeline for GUI data construction. And it shows promising results on a wide range of benchmarks.

**Other Comments Or Suggestions:**

The paper is well-written and structured logically.

**Other Strengths And Weaknesses:**

Strengths:
- The performance is outstanding across multiple benchmarks.

Weaknesses:
- Recently, many co-current works have proposed similar unified model architectures, such as UI-TARS, OS-ATLAS, ShowUI, and CogAgent-9B. Could the authors compare their approach with these works, particularly in terms of data? The proposed method in this paper appears to be one of the best in balancing data utilization efficiency and model performance.

**Questions For Authors:**

The paper is well-prepared, and I have no further questions.

**Relation To Broader Scientific Literature:**

Unified Vision and Action Model: AGUVIS extends research on GUI understanding through vision models, moving beyond traditional approaches that rely on accessibility trees or HTML (like WebGPT, Mind2Web).
CoT on GUI Domain: It also incorporates inner monologue techniques to enhance reasoning capabilities in multimodal contexts.

**Theoretical Claims:**

This is not a theoretical paper.

---

> ### Author Rebuttal · Authors · 2025-04-01
>
> Thank you for your thoughtful review of our AGUVIS paper. We appreciate your recognition of our comprehensive roadmap for developing a pure-vision GUI agent, particularly our data curation approach and training strategies. Your positive assessment of our experimental validations across multiple benchmarks is encouraging.
>
>
> > Q1. Recently, many co-current works have proposed similar unified model architectures, such as UI-TARS, OS-ATLAS, ShowUI, and CogAgent-9B. Could the authors compare their approach with these works, particularly in terms of data? The proposed method in this paper appears to be one of the best in balancing data utilization efficiency and model performance.
>
> We're pleased to see the growing interest in GUI agent research through concurrent works like UI-TARS, OS-ATLAS, ShowUI, and CogAgent-9B[4]. In comparison with these impressive efforts, we believe that AGUVIS makes a unique and complementary data contribution to the GUI agent community in several important ways:
>
> - **Reasoning via Inner Monologue:** A defining feature of AGUVIS is the use of reasoning via inner monologue, which is not present in many concurrent works such as OS-ATLAS, ShowUI, and CogAgent-9B. This inner monologue allows AGUVIS to perform reasoning during interaction, which is crucial for the effective handling of complex tasks across multiple platforms (mobile, desktop, web). As shown in Section 4.3, AGUVIS achieves strong generalization by using a unified action space after inner monologue reasoning, enabling knowledge transfer across diverse environments. Additionally, the inner monologue surprisingly enhances GUI grounding performance, as detailed in Section 4.2 and Appendix E.1.2, as well as in our response to [Reviewer cBz9 Q1](https://openreview.net/forum?id=PlihOwfx4r&noteId=AzjUJ6TwN5). We believe that these contributions jointly underpin our balance between data utilization efficiency and model performance.
> - **Open-Source Data Collection and Pipeline:** While recent concurrent works, such as OS-ATLAS and ShowUI, focus on grounding-centric training data, and more recent UI-TARS has made significant strides in leveraging in-house human-annotated trajectories with reasoning, AGUVIS offers a unique advantage with its large-scale, open-source data collection. Our dataset collection not only includes both unified grounding and trajectory annotations but also integrates reasoning into the data pipeline. This transparency and open-source nature of our data collection make AGUVIS a valuable resource for the community to build upon and extend our work more easily.
>
> We greatly appreciate your recognition of AGUVIS as one of the leading approaches in balancing data efficiency and model performance. We believe that our open-source data and the novel inner monologue reasoning offer complementary contributions that will drive continued progress in the development of autonomous GUI agents.

---

### Official Review · Reviewer_cBz9 · 2025-03-14

**Overall Recommendation:** 4

**Summary:**

This paper introduces AGUVIS, a vision-based UI agent designed to operate across diverse digital platforms. The authors collect data from existing resources and do some essential augmentation. They then leverage a vision-language model to train AGUVIS in two stages, grounding and planning, to improve interaction capabilities. The framework is evaluated on multiple datasets, including grounding, offline agent and online agent benchmarks, demonstrating strong performance across various GUI benchmarks.

**Claims And Evidence:**

The claims are adequately supported with sufficient experimental results.

**Essential References Not Discussed:**

N/A

**Experimental Designs Or Analyses:**

In Section 4.2 and Table 6, the authors state that Inner Monologue benefits both grounding and planning. However, only the planning stage (Stage 2) incorporates Inner Monologue, while the grounding stage (Stage 1) only relies on existing datasets without such augmentation. So I wonder how Inner Monologue contributes to performance gains in grounding? I notice that the authors provide some explanation in Appendix E.1.2 that "This is mainly because the low-level instructions of inner monologue act as atomic instruction and grounding action pairs, also enhancing the grounding ability of our GUI agents." However, it's kind of hard to understand this explanation. It would be helpful if the authors could elaborate further on this point.

**Methods And Evaluation Criteria:**

The method is simple and straightforward. The authors use GPT-4o to "translate" GUI actions into natural language, which helps agent understanding during fine-tuning. They then apply standard fine-tuning on the aggregated grounding data and augmented planning trajectories. Essentially, they do not create a new dataset or propose a novel framework, so the methodological novelty may seem limited. However, significant engineering effort is also involved, such as aggregating datasets from various sources, standardizing formats, validating through human studies, etc. Also, the authors open-source their trained models and datasets, which could benefit the academic community. These altogether could build up sufficient contribution for the paper overall.

The evaluation benchmarks are comprehensive, including grounding, offline GUI agent and online GUI agent evaluation.

**Other Comments Or Suggestions:**

N/A

**Other Strengths And Weaknesses:**

See above.

**Questions For Authors:**

1. Could the authors further clarify the difference between the self-plan and enforced plan settings? I reviewed the prompt templates on pages 27 and 28 but couldn’t identify any differences between them. Could the authors provide an explanation?

2. Regarding the proposed Aguvis model, does it generate the next step iteratively, or does it first generate an overall plan and then generate each step iteratively? I assume it's the first one, but would like to confirm.

**Relation To Broader Scientific Literature:**

The open-sourced dataset and model checkpoints are helpful for future academic research.

**Theoretical Claims:**

N/A. The paper does not include much theoretical claim. (Not a weakness)

---

> ### Author Rebuttal · Authors · 2025-04-01
>
> We sincerely appreciate your thorough review and positive assessment of our work. We're particularly encouraged by your recognition of our comprehensive evaluation benchmarks, the engineering effort involved in dataset aggregation and standardization, and the value of our open-sourced models and datasets to the research community.
>
> Regarding your questions and comments:
>
> > **Q1: On Inner Monologue's Contribution to Grounding Performance**
>
> Thank you for this insightful question. You correctly observed that while Inner Monologue is introduced in Stage 2, it also enhances grounding performance, which might seem counterintuitive. Let us clarify:
> In standard trajectories without inner monologue, the data structure is:
> > (High-level goal G, observation $o_1$, action $a_1$, observation $o_2$, action $a_2$, ...)
>
> When we augment with inner monologue, we introduce low-level instructions, transforming it to:
>
> > (High-level goal G, observation $o_1$, low-level instruction $a_1^{inst}$, action $a_1$, observation $o_2$, low-level instruction $a_2^{inst}$, action $a_2$, ...)
>
>
> This transformation creates high-quality explicit instruction-action pairs ($o_i$, $a_i^{inst}$, $a_i$) within each step, essentially embedding "grounding examples" throughout the trajectory. The model learns to:
> 1. Interpret high-level goals into precise low-level instructions
> 2. Ground these instructions to specific UI elements
> 3. Generate appropriate actions
>
> As shown in Table 6, removing inner monologue reduces performance on ScreenSpot from 84.4% to 79.3% and also has a strong impact on AndroidControl Low-Level tasks (80.5% → 69.1%). This suggests that the ability to decompose tasks into explicit low-level instructions significantly improves grounding precision.
>
> > **Q2: On Self-Plan vs. Enforced Plan Settings**
>
> Thank you for pointing out this confusion. As shown in Appendix E.2.1 and Figure, the difference between these settings lies in how we prompt the model in response:
>
> **Self-Plan Setting:**
>
> ```
> <|im_start|>assistant<|recipient|>
> [The model decides whether to plan first with all or directly execute actions with os]
> ```
>
>
> In this setting, the model autonomously determines whether to generate planning thoughts based on task complexity. For simple tasks like "Click the 'Buy' button," it might directly output:
>
> ```
> <|im_start|>user
> Click the 'Buy' button.
> <|im_end|>
> <|im_start|>assistant<|recipient|>os
> pyautogui.click(0.34, 0.45)
> <|im_end|>
> ```
>
> While for some implicit tasks, it might choose to plan first:
>
> ```
> <|im_start|>user
> Send current webpage.
> <|im_end|>
> <|im_start|>assistant<|recipient|>all
> Thought: To share the current page, I need to find and click the share icon, which is typically represented by a network or link symbol. This icon is usually located in the browser's toolbar or menu.\nAction: Click the share icon in the browser to share the current page.
> <|im_end|>
> <|im_start|>assistant<|recipient|>os
> pyautogui.click(0.34, 0.45)
> <|im_end|>
> ```
>
> **Enforced Plan Setting:**
>
> ```
> <|im_start|>assistant<|recipient|>all
> Thought: [The model is forced to generate planning thoughts before actions]
> ```
>
>
> The enforced plan setting explicitly requires the model to engage in high-level reasoning before taking actions. As noted in our error analysis (Section 4.5), this enforced planning resolves approximately 20% of grounding errors by encouraging the model to carefully consider the task context, potential ambiguities, and available UI elements before committing to action.
>
> We will further clarify this part in our revised manuscript. Thanks for helping improve our work!
>
>
> > **Q3: On AGUVIS Model's Generation Approach**
>
> Yes, AGUVIS generates the next step iteratively rather than first generating an overall plan and then executing steps. At each time step, given the current observation and task history, the model:
>
> 1. Generates thoughts about the current state in relation to the goal
> 2. Determines the appropriate next action
> 3. Executes the action and receives a new observation
> 4. Repeats the process until task completion
>
> This iterative approach allows AGUVIS to adapt to changing UI states and unexpected outcomes during task execution, rather than rigidly following a predetermined plan.

---

### Decision · Program_Chairs · 2025-05-01

**Decision:**

Accept (poster)

**Comment:**

This paper introduces Aguvis, a novel, purely vision-based framework for autonomous GUI interaction across different platforms. Key contributions include a unified agent architecture operating directly on screen images, the integration of structured reasoning via "inner monologue," a large-scale open-source dataset (Aguvis Data Collection) with multimodal annotations, and a two-stage training pipeline separating grounding and planning. The authors demonstrate state-of-the-art performance on various offline and online benchmarks without reliance on closed-source models during execution.

All reviewers recognize the paper's strengths, particularly the impressive performance achieved across a comprehensive set of benchmarks (cBz9, VD9H, X3pb, zR1B). The open-sourcing of the dataset, model, and training pipeline is consistently highlighted as a valuable contribution to the community (cBz9, VD9H, zR1B). The unified vision-based approach and the incorporation of inner monologue for reasoning are also seen as positive aspects (cBz9, VD9H). The engineering effort in data collection and standardization was acknowledged (cBz9).

Initial concerns included the perceived novelty of the training pipeline compared to concurrent work (VD9H, X3pb), clarification needed on the "fully autonomous without closed-source models" claim given the use of GPT-4o in data annotation (X3pb), understanding the mechanism by which inner monologue improved grounding (cBz9), and requests for additional comparisons or evaluations (VD9H, X3pb, zR1B).

The authors provided thorough rebuttals addressing these points. They clarified that the core innovation lies in the integration of inner monologue within their unified framework enabling cross-platform generalization, explained the grounding benefits, detailed the autonomy claim (runtime independence from closed-source models), provided comparisons with concurrent work focusing on data contributions, and presented strong additional results on the AITZ benchmark as requested (X3pb). Reviewers acknowledged the rebuttals, with Reviewer X3pb raising their score and Reviewer zR1B confirming their leaning towards acceptance post-rebuttal.

In conclusion, the paper presents a nice contribution towards general-purpose, vision-based GUI agents. It demonstrates good empirical results on diverse benchmarks and provides valuable open-source resources. The authors effectively addressed reviewers' concerns in their rebuttal, strengthening the paper and leading to a clearer consensus towards acceptance among the reviewers. The work's performance, open contribution, and novel integration of reasoning make it suitable for publication.

I suggest the authors to discuss these works in the revision as they resemble some of the earlier works for incorporating planning trajectories (like inner monologue) in UI grounding:

Devil's Advocate: Anticipatory Reflection for LLM Agents, Wang et al., EMNLP Findings 2024

MUG: Interactive Multimodal Grounding on User Interfaces, Li et al., EACL 2024

A Zero-Shot Language Agent for Computer Control with Structured Reflection, Li, et al., EMNLP 2023